# Measuring Variable Importance in Heterogeneous Treatment Effects with Confidence

**Joseph Paillard** [1 2]  **Angel Reyero Lobo** [2 3]  **Vitaliy Kolodyazhniy** [1]  **Bertrand Thirion** [2]  **Denis A. Engemann** [1]

## Abstract

Causal machine learning holds promise for estimating individual treatment effects from complex data. For successful real-world applications of machine learning methods, it is of paramount importance to obtain reliable insights into which variables drive heterogeneity in the response to treatment. We propose `PermuCATE`, an algorithm based on the Conditional Permutation Importance (CPI) method, for statistically rigorous global variable importance assessment in the estimation of the Conditional Average Treatment Effect (CATE). Theoretical analysis of the finite sample regime and empirical studies show that `PermuCATE` has lower variance than the Leave-One-Covariate-Out (LOCO) reference method and provides a reliable measure of variable importance. This property increases statistical power, which is crucial for causal inference in the limited-data regime common to biomedical applications. We empirically demonstrate the benefits of `PermuCATE` in simulated and real-world health datasets, including settings with up to hundreds of correlated variables.

## 1. Introduction

The ever growing volume of high-dimensional biomedical data holds promise to facilitate the discovery of novel biomarkers that can predict individual treatment success or risk of adverse events (Pang et al., 2023; Wu et al., 2020; Kong et al., 2022). Transforming complex biological signals into actionable variables, therefore, plays a crucial role for increasing the efficiency of biomedical research

and development (Frank & Hargreaves, 2003; Hartl et al., 2021). Machine Learning (ML) promises to improve the prediction of biomedical and clinical outcomes from high-dimensional, high-throughput data in various biomedical applications due to its ability to learn complex functions and representations from heterogeneous and correlated data, such as those encountered in e.g. imaging, genomics or transcriptomics data (Le Goallec et al., 2022; Parrish et al., 2024; DeGroat et al., 2023). Regardless, their notorious lack of interpretability hinders the applicability of ML models for generating scientific and clinical insights, motivating efforts for new methods for estimating variable importance within the field of interpretable ML (Molnar, 2020; Biecek & Burzykowski, 2021). Most studies so far have focused on classical ML models like random forests (Strobl et al., 2008), which are well-suited for tabular data with few variables (Grinsztajn et al., 2022). A more recent line of work, exposed to complex life-science data, has therefore emphasised the importance of high-capacity ML algorithms and proposed methods combining high-capacity ML algorithms with inference algorithms backed by statistical guarantees (Mi et al., 2021; Williamson et al., 2023; Lei et al., 2018; Chamma et al., 2024).

Despite significant efforts and advances on methods for interpretability, the utility of ML models remains limited as biomedical data are often of interventional nature. Therefore, such datasets are typically analyzed using biostatistical methods appropriate for causal inference and accepted by regulatory agencies. Over the past decade, findings from the causal inference and potential outcomes literature (Imbens & Rubin, 2015) have been adapted to propose the estimand framework (Akacha et al., 2017; Ratitch et al., 2020), a flexible statistical-modelling framework useful for causal analysis of clinical data. It provides formal concepts and draws a conceptual distinction between the quantity of interest and the estimation method. This enables a formal language that can bridge the gap between classical biostatistics and ML, hence, pave the way for causal ML in biomedical applications (Feuerriegel et al., 2024; Sanchez et al., 2022; Doutreligne & Varoquaux, 2025). When studying heterogeneity of treatment effects using ML, explainability with statistical guarantees is crucial for uncovering useful predictive biomarkers as ad-hoc methodology may increase

---

[1]Roche Pharma Research & Early Development, Roche Innovation Center Basel, F. Hoffmann-La Roche Ltd, Basel, Switzerland; [2]Université Paris-Saclay, Inria, CEA, Palaiseau, France; [3]Institut de Mathematiques de Toulouse, UMR5219 Université de Toulouse, France. Correspondence to: Joseph Paillard <joseph.paillard@roche.com>.

*Proceedings of the $42^{nd}$ International Conference on Machine Learning*, Vancouver, Canada. PMLR 267, 2025. Copyright 2025 by the author(s).

the false positives rate as well as the false negative rate.

Taken together, in the context of present and future biomedical applications, ML methods must address both the complexity of multimodal high-dimensional biological data and support the research questions that arise from interventional studies with estimands such as average treatment effects (ATE) and conditional average treatment effects (CATE). In the causal setting, it remains unclear whether variable importance can be measured with intact statistical guarantees.

**Contributions**. In this work we study algorithms for estimating the importance of variables with statistical guarantees in the context of interventional data and CATE.

- We propose a new variable importance inference procedure for heterogeneous treatment effects termed `PermuCATE`, which generalizes Conditional Permutation Importance (`CPI`) (Chamma et al., 2023). We put it in perspective with the established leave-one-covariate-out (`LOCO`) method (Williamson et al., 2023; Lei et al., 2018).

- We develop a theoretical analysis of the finite sample variance for both `PermuCATE` and `LOCO`, focusing on the impact of estimation error in finite samples.

- Through experiments on simulated and real-world data, using various causal ML models, we show how `PermuCATE` achieves lower variance and higher statistical power due to the influence of finite sample estimation error, an effect that is more pronounced in small sample sizes or when using complex deep learning CATE estimators.

All proofs and additional experiments are given in appendix.

## 2. Related Work and Setting

### 2.1. Related Work

We first review prior art on variable importance in standard prediction problems and, subsequently, their applications to heterogeneous treatment effects. Previous works have analysed (Verdinelli & Wasserman, 2023) and benchmarked (Chamma et al., 2023) a wealth of procedures and statistical methods for variable importance estimation (Strobl et al., 2008; Nguyen et al., 2022; Watson & Wright, 2021; Covert et al., 2020; Candes et al., 2018; Louppe et al., 2013). Out of those approaches for standard prediction problems, `LOCO` (Williamson et al., 2023) and `CPI` (Chamma et al., 2023) satisfied the requirements of being model-agnostic (therefore compatible with high-capacity prediction models), offering statistical guarantees for controlling type-1 error, having a realistic computation cost and being interpretable (Chamma et al., 2023). In addition to enabling variable selection, similar to methods like

knockoffs, both approaches measure the influence of each variable on the output of a (possibly complex) model by estimating the total Sobol index. This well-known quantity can be interpreted as the loss increase when a variable is removed from the input of a predictive model. It has also been interpreted as a generalized form of ANOVA (Sobol, 2001; Williamson et al., 2021).

Prior research also studied the problem of variable importance for CATE estimation in the case of causal forest estimators (Bénard & Josse, 2023) or Bayesian Additive Trees (BART) (Chipman et al., 2010). For biomarker detection in the CATE context, Sechidis et al. have investigated knockoff procedures with LASSO models and causal forests. However, these approaches limit the choice of prediction models. Based on the behavior of random forests and knockoffs in standard prediction problems (Chamma et al., 2023), we would expect that both approaches encounter limits when it comes to nonlinear effects in high-dimensional data with high correlations. Another line of work focused on interpretability using attribution methods such as Shapley values or integrated gradients (Crabbé et al., 2022). While such methods are often good at ranking important variables, they do not provide type-1 error control. This is particularly true for Shapley values, a measure of importance based on an additivity axiom, meaning that estimated importance tends to be smeared over correlated variables. This property necessarily leads to false positives in the presence of correlation. Hines et al. (2022) have developed a `LOCO` procedure in concert with the potential-outcomes feasible risk, hence, extending a statistically grounded variable-importance method to the CATE context.

As pointed for standard prediction problems (Chamma et al., 2023; 2024), in contrast with permutation methods, `LOCO` is a refitting procedure requiring a full model fit per variable of interest. In the CATE context, the required amount of computation is therefore expected to be higher, depending on the meta-learner and the choice of feasible risk. It is thus a question of interest if `CPI` can be further developed to support variable importance in CATE estimation. Finally, a neglected aspect of previous studies of `LOCO` and `CPI` is the impact of estimation error that stems from the refitting and the covariate estimation, respectively.

The following sections present a formal treatment of the related work and prepare the ground for our contributions.

### 2.2. Problem setting

Capital letters denote random variables, and lowercase letters denote their realizations. We consider the potential-outcomes framework with a set of observations $Z_i = (X_i, A_i, Y_i(A_i))$. Here, $X_i \in \mathbb{R}^d$ denotes the set of covariates for subject $i$, $A \in \{0, 1\}$ indicates a binary treatment assignment, and $Y_i(A_i)$ represents the observed outcome.

We focus on the estimation of the Conditional Average Treatment Effect $\tau(x) = \mathbb{E}[Y(1) - Y(0)|X = x]$, which captures the expected difference between the outcome under treatment ($A_i = 1$) and control ($A_i = 0$) for a subject with covariates $X = x$. Unlike classical ML problems, this function depends on potential outcomes, which are unobserved data since each subject is assigned to either the treatment or control group. This leads to fundamental challenges in the estimation procedure of this function as well as in scoring the candidate estimate since the ground truth is not readily available. For a background on CATE estimation and feasible risks, we recommend the work by Kennedy (2023) and by Doutreligne & Varoquaux (2025), respectively.

**Leave-One-Covariate-Out (LOCO) for CATE:** Unlike most efforts centered on the estimation of the CATE, in this work, we focus on measuring the importance of the different covariates $X^j, j = 1, \cdots, d$ for CATE prediction. In their work on variable importance for general ML problems, (Williamson et al., 2023) analysed in detail the Leave-One-Covariate-Out (LOCO) approach for estimating the importance of a variable (or group of variables) as the decrease in performance between an estimator fitted on the full set of covariates $X$ and the subset of covariates excluding the $j^{th}$, that we denote $X^{-j}$. This approach was then adapted to the causal framework (Hines et al., 2022), providing a variable importance measure (VIM) given a risk $R$ that measures the predictiveness of a CATE estimator,

$$\Psi_{LOCO}^j = R\left(\tau^{-j}, X^{-j}, A, Y\right) - R\left(\tau, X, A, Y\right) \quad (1)$$

where $\tau^{-j}$ being the CATE estimator fitted on the subset of covariates. In practice, CATE estimators are fitted on a training set, and $\hat{\Psi}_{LOCO}^j$ is evaluated on a disjoint test set.

**Feasible Risks for Causal Inference:** The initial formulation of the variable importance measure by Williamson et al. 2023 allows for the choice of different metrics to evaluate the estimator. However, the extension of this framework encounters the fundamental problem in causal inference: the oracle CATE function is not accessible in practice (Holland, 1986). Thus, applying ML metrics commonly used for empirical risk minimization in prediction tasks, such as the precision of estimating heterogeneous effects (PEHE), which is the Mean-Squared Error (MSE) in estimating the CATE, cannot be computed. To overcome this hurdle, a feasible risk that can be computed from real data should be used to provide practical methods. Hines et al. 2022 for instance used the PO-risk: the mean squared error between the CATE prediction and the pseudo-outcome given by

$$\varphi(z) = \frac{(y - \mu_a(x))(a - \pi(x))}{\pi(x)(1 - \pi(x))} + \mu_1(x) - \mu_0(x), \quad (2)$$

where $\mu_0$ and $\mu_1$ denote the response functions: $\mu_a(x) = \mathbb{E}[Y|X = x, A = a]$ and $\pi(x) = \mathbb{E}[A = 1|X = x]$ is the propensity score. Alternatively, a recent review of causal risks (Doutreligne & Varoquaux, 2025) for model selection in causal inference suggested using the R-risk given by

$$((Y - m(X)) - (A - \pi(X))\tau(X))^2 \quad (3)$$

where $m(X) = \pi(X)\mu_1(X) + (1 - \pi(X))\mu_0(X)$. Both risks are consistent with the oracle PEHE (proof in Appendix A.2) when the ground-truth nuisance functions are used. Additionally, for variable importance, they empirically produce the same results (Figure S1). We used the PO-risk for the rest of this work as it is easily comparable to the oracle PEHE, has been used in previous publication and is easy to compute.

**Conditional Permutation Importance (CPI):** Another line of work on conditional VIM has introduced the Conditional Permutation Importance (CPI) measure (Chamma et al., 2023). Instead of re-fitting the estimator on a subset of covariates, this method involves sampling the given subset from the conditional distribution and evaluating the impact on the loss. This provides a second estimator of variable importance, denoted $\Psi_{CPI}^j$ described in the case of CATE in equation 4. Similar to LOCO, this approach provides confidence intervals, is model-agnostic, and enjoys type-I error control. However, estimating the conditional density as required by CPI is often less error-prone than estimating a sub-model in LOCO since in many practical situations, the relationships among the covariates are simpler than the mechanisms linking the covariates to the outcome as discussed in Remark 3.3. Finally, both approaches present similarities, among which, converging asymptotically to the total Sobol index (Sobol, 2001).

**Computational efficiency:** Unlike LOCO, CPI reuses the same estimator when predicting from the perturbed set of covariates. This reduces both the computational cost and the error that comes from stochastic models optimization. It is worth noting that in the work from Hines et al. 2022 adapting LOCO for CATE estimation, only the second stage of the DR-learner is re-fitted on the subset of covariates: $\tau_{-j} = \mathbb{E}[\varphi(z)|X^{-j} = x^{-j}]$. By contrast, the pseudo-outcomes $\varphi(z)$ (cf Equation 2), are obtained from the full set of covariates. Bypassing the intermediate estimation of the nuisance functions, this trick mitigates the extra computational cost incurred by LOCO when applied to causal estimators as opposed to regular ML problems and avoids violating the unconfoundedness assumption when estimating the CATE with missing covariates. However, in scenarios where this final estimation step requires complex models, such as super-learners (Laan et al., 2007) or deep neural networks (Curth & Schaar, 2021) the computational burden can become intractable.

## 3. `PermuCATE` algorithm

The method presented in this section aims at identifying important variable for CATE prediction. While similarly to LOCO, it is agnostic to the meta-learning approach used to estimated the CATE, we used the DR-learner (Kennedy, 2023). This choice is motivated by its favorable convergence properties and the fact that the scoring step requires deriving pseudo-outcomes explicitly, which limits the computational benefits of simpler and more direct meta-learning approaches such as the S- or T-learner.

---

**Algorithm 1** Conditional Permutation Importance for CATE

**Input**: $D_{train}, D_{test}$ two independent sets of observations $Z_i = (X_i, A_i, Y_i), X_i \in \mathbb{R}^d$
**Parameter**: Response functions: $\hat{\mu}_0, \hat{\mu}_1$; Propensity: $\hat{\pi}$; CATE: $\hat{\tau}$; Covariate predictor: $\hat{\nu}(\cdot)$, feasible risk: $R(\cdot)$, Number of permutations: $P$
**Output**: Importance estimate $\hat{\Psi} \in \mathbb{R}^d$

1: Using $D_{train}$
2: Fit $\hat{\mu}_0, \hat{\mu}_1, \hat{\pi}, \hat{\tau}$ // CV can be used for DR-learner
3: **for** $j = 1, \cdots, d$ **do**
4:      Fit $\hat{\nu}_j(\cdot) \leftarrow \hat{\mathbb{E}}_n[X^j | X^{-j}]$
5: **end for**
6: Using $D_{test}$
7: **for** $j = 1, \cdots, d$ **do**
8:      $\hat{\nu}_j \leftarrow \hat{\nu}_j(X^{-j})$
9:      $r_j \leftarrow X^j - \hat{\nu}_j$
10:      **for** $k = 1, \cdots, P$ **do**
11:          $\widetilde{r}_j \leftarrow \text{shuffle}(r_j)$
12:          $X_{P,j} \leftarrow [X_1, \cdots, X_{j-1}, \hat{\nu}_j + \widetilde{r}_j, \cdots, X_d]$
13:          $\hat{\Psi}_k^j \leftarrow (R(\hat{\tau}(X_{P,j}), Y, A) - R(\hat{\tau}(X), Y, A))/2$
14:      **end for**
15:      $\hat{\Psi}^j \leftarrow \frac{1}{P} \sum_{k=1}^{P} \hat{\Psi}_k^j$
16: **end for**
17: **return** $[\hat{\Psi}^1, \cdots, \hat{\Psi}^d]$

---

LOCO estimates the importance of a given variable by refitting a model on a subset of covariates, whereas permutation-based approaches reuse the same model to predict from a perturbed design matrix. In the traditional permutation importance approach (Breiman, 2001), the perturbed matrix differs from the original by a permutation of the studied covariate. That approach is naive in the sense that it does not account for correlation, leading to uncontrolled type-1 errors; CPI, a refined version mitigates this issue (Chamma et al., 2023). The key idea is to sample from the conditional distribution $p(X^j | X^{(-j)})$ which requires the following assumption.

**Assumption 3.1.** There exists a function $\nu_j$ such that $X^j = \nu_j(X^{(-j)}) + \varepsilon_j$ with $\varepsilon_j \perp\!\!\!\perp X^{(-j)}$ and $\mathbb{E}[\varepsilon_j] = 0$.

Sampling from the conditional can thus be achieved by

estimating $\nu_j$ with a consistent estimator $\hat{\nu}_j$ (see Proposition 3.3 from Reyero Lobo et al. (2025)). Then, the residual information $r_j = X^j - \hat{\nu}_j$ can used to construct a perturbed version of the $j^{th}$ covariate: $\hat{\nu}_j + \widetilde{r}_j$ where $\widetilde{r}_j = \text{shuffle}(r_j)$ is a permutation of the residual.

**Proposition 3.2.** *Assume the consistency of the estimators $\hat{\tau}$ and Assumption 3.1.* `PermuCATE` *consistently estimates $\mathbb{E}\left[\mathbb{V}\left[\tau(X)|X^{(-j)}\right]\right]$, the quantity known as the total Sobol index.*
*The proof is provided in subsection A.3.*

*Remark* 3.3. Assumption 3.1 amounts to assuming that the relationships between the covariates within $X$ are simpler than the link between the CATE, $\tau$ and the input $X$. Consequently, estimating the conditional density is simpler than estimating the outcome, akin to the '*model-X*' knockoff framework presented in Candes et al. 2018. Numerous examples illustrate the utility of this setting, such as in the study of biologically complex diseases using multiple variables like single nucleotide polymorphisms (SNPs) where the relationship between covariates is much simpler than their association with clinical outcomes. Similarly, in clinical trials, the covariates are collected concurrently (e.g. at baseline) but the outcome of interest is measured at a later time. Furthermore, the amount of unlabelled data often exceeds the number of available labels, especially in clinical settings where labelling can be costly.

To check if Assumption 3.1 is realistic, we conducted computational experiments via simulations and analysis of real-world medical data. We focused on the IHDP benchmark which uses real-world data on infant-health development. The dataset has non-Gaussian continuous features and imbalanced categorical (up to four categories) features. Because the ground truth for the conditional distribution is not available, the validity of Assumption 3.1 cannot be assessed. In addition to this benchmark, we propose an additional experiment varying the complexity of the conditional distributions. For this purpose, we use simulations inspired by Hollmann et al. 2025 and sample the covariates using a latent variable model. We first sampled latent variables from mixtures of Gaussians and then generated observed covariates through a non-linear transformation with interaction terms of the latent variables. This resulted in non-Gaussian covariates with complex conditional distributions as shown in Figure S5.

Adapting this idea to the context of CATE estimation, given a causal risk $R$, the following measure of variable importance can be obtained:

$$\Psi_{CPI}^j = \frac{1}{2}\left(R\left(\tau, X_{P,j}, A, Y\right) - R\left(\tau, X, A, Y\right)\right) \quad (4)$$

where $X_{P,j} = [X_1, \cdots, X_{j-1}, \nu_j + \widetilde{r}_j, \cdots, X_d]$. `PermuCATE` (Algorithm 1) describes the procedure to obtain $\Psi_{CPI}^j$. Previous works focusing on predictive models have shown how a cross-fitting approach allows to obtain a

valid estimate of the desired quantity with a control of the type-I error (Williamson et al., 2023; Verdinelli & Wasserman, 2023; Chamma et al., 2023; Reyero Lobo et al., 2025).

**Properties** The following paragraphs compare the properties of the LOCO and PermuCATE estimators in the context of finite training sets $\mathcal{D}_{\text{train}}$, to highlight variance-inducing error terms from finite sample estimations. A key advantage of PermuCATE is that it allows the use of an expressive model for the main function estimation, such as a deep neural network, without incurring the high cost for refitting the full model, provided that a more parsimonious model is used for covariate prediction. However, in finite sample regimes, this estimation of $\hat{\nu}_j$ on a fixed train set $D_{\text{train}}$ introduces variance. We denote $||\cdot||^2$ is the l2-norm given the fixed training set, $\mathcal{D}_{\text{train}}$.

**Proposition 3.4.** *Under Assumption 1, for a consistent CATE estimator $\hat{\tau}$, the finite sample bias in the estimation of the importance for the $j^{th}$ covariate is given by,*

$$\mathbb{E}\left[\widehat{\Psi}^j_{\text{CPI}}|\mathcal{D}_{\text{train}}\right] - \Psi^*_j =$$

$$\mathbb{E}[(\hat{\tau}(X_{P,j}) - \hat{\tau}(\hat{X}_{P,j}))^2|\mathcal{D}_{\text{train}}] + O_P(\mathbb{E}[\tau - \hat{\tau}|\mathcal{D}_{\text{train}}])$$

*where $\Psi^*_j = \mathbb{E}\left[\mathbb{V}\left[\tau(X)|X^{(-j)}\right]\right]$ is the targeted total Sobol index.*

*Remark: Although the first term involves the CATE estimate $\hat{\tau}$, it does not include the estimation error, thus making it more robust to model misspecifications. The estimation error appears only in the second, linear term, instead of the mean-squared error (MSE) present for LOCO. The proof for that relies on the fact that MSE terms cancel out because $\mathbb{E}[(\tau(X_{P,j}) - \hat{\tau}(X_{P,j}))^2|\mathcal{D}_{\text{train}}] = \mathbb{E}[(\tau(X) - \hat{\tau}(X))^2|\mathcal{D}_{\text{train}}]$, since by construction $X_{P,j} \overset{\text{i.i.d.}}{\sim} X$.*

*Corollary: Under the additional assumption that $\hat{\tau}$ is Lipschitz continuous and $\hat{\nu}$ is consistent, the bias term becomes*

$$O_P(||\nu_j - \hat{\nu}_j||^2) + O_P(\mathbb{E}[\tau - \hat{\tau}]) \tag{5}$$

*Proof in Appendix A.4.*

In line with Proposition 3.2, it can be observed that, given a consistent estimator of $\nu_j$, $\widehat{\Psi}^j_{\text{CPI}}$ is a consistent estimator of the total Sobol index. It also shows that the error comes from estimating $\hat{\nu}_j$ To better understand the impact of estimation error on variance, consider a linear setting with Gaussian covariates where the CATE is defined as $\tau(X) = \beta X$, with a finite training sample and an infinitely large testing sample, the variance of the PermuCATE estimator is given by

$$\text{var}\left(\frac{1}{2}\mathbb{E}\left[\widehat{\Psi}^j_{\text{CPI}}|\mathcal{D}_{\text{train}}\right]\right) = \text{var}(\widehat{\beta}^2_j||\Delta\widehat{\gamma}||^2_{\Sigma_{-j}}) \tag{6}$$

where $\Delta\widehat{\gamma} = \gamma - \widehat{\gamma}$ is the error when estimating the coefficients of the function $\nu_j(X^{-j}) = \gamma X^{-j}$ on the training set,

$\mathcal{D}_{\text{train}}$ and $\Sigma_{-j}$ is the covariance matrix of $X^{-j}$ (proof in Appendix A.8). Consequently, Equation (6) demonstrates that the variance of PermuCATE is mainly influenced by the error in estimating the conditional distribution via $\hat{\nu}$ but not on the error in $\hat{\tau}$. Concerning the LOCO approach, the iterative refitting of the model on the subsets of covariates $X^{-j}$ leads to fluctuations due to the estimation of $\hat{\tau}$ and introduces a bias term associated with the CATE prediction.

**Proposition 3.5.** *For a consistent CATE estimator $\hat{\tau}$, the finite sample estimation error of the importance for the $j^{th}$ covariate is*

$$\mathbb{E}\left[\widehat{\Psi}^j_{\text{LOCO}}|\mathcal{D}_{\text{train}}\right] - \Psi^*_j =$$

$$O_P\left(||\tau_{-j} - \widehat{\tau}_{-j}||^2 - ||\tau - \widehat{\tau}||^2\right) \tag{7}$$

*Proof in Appendix A.5.*

Denoting $\Delta\hat{\beta}$ and $\Delta\hat{\beta}_{-j}$ the difference between the estimated and true coefficients for respectively $\tau(X) = \beta X$ and the sub-model $\tau_{-j}$. In the linear scenario as presented above the finite sample variance of the LOCO estimator is then

$$\text{var}\left(\mathbb{E}\left[\widehat{\Psi}^j_{\text{LOCO}}\Big|\mathcal{D}_{\text{train}}\right]\right) = \text{var}\left(||\Delta\hat{\beta}_{(-j)}||^2_{\Sigma_{-j}}\right)$$

$$+ \text{var}\left(||\Delta\hat{\beta}||^2_{\Sigma}\right) \tag{8}$$

In the above equations, which present the different bias and variances in a linear setting assuming independent training samples, a key distinction emerges between the two approaches. The variance of LOCO depends on the sum of the error in estimating the CATE $\hat{\tau}$ and its sub-model $\hat{\tau}_j$. In contrast, for PermuCATE it depends on the error in estimating the conditional distribution (through the error in $\hat{\nu}_j$). As discussed in Remark 3.3, the latter prediction problem is arguably easier to solve. Supplementary Figure S2 provides an illustration of this comparison.

Therefore, we investigated the variance behavior of both approaches using finite-sample simulations and datasets. The results in the following section will show empirically that the error term in LOCO can lead to increased variance and reduced statistical power.

## 4. Experiments

**Datasets:** This work studies three different simulation scenarios and a real-world dataset. In the paper, we refer to high-dimensional data as it is commonly understood in the field of medical research, assuming that prior dimensionality reduction has been performed, thereby limiting the number of variables to at most a few hundred. Scenarios involving much larger dimensions are not addressed here due to the high correlation between covariates, which leads to vanishing importance and a consequent lack of statistical power.

However, our proposed approach is compatible with grouping strategies (Chamma et al., 2024), as illustrated in the Figure S3. This allows for inference on the importance of groups of covariates rather than individual covariates, helping to mitigate the challenges posed by highly correlated variables and facilitating the handling of higher-dimensional data when they can be grouped. The following list presents an overview of the datasets, further details are provided in the Appendix A.10.

- The Low Dimensional (*LD*) dataset is the data generating process 3 from the work of Hines et al. 2022. It is used as a benchmark to compare the proposed approach to their baseline method. It consists of six covariates, each pair (($X_1, X_2$), ($X_3, X_4$), ($X_5, X_6$)) being correlated ($\rho = 0.5$). Three of these covariates ($X_1, X_2, X_3$) are linked to the nuisance functions and CATE through linear functions. The analytical importance for these variables is respectively $\Psi^1 = 0.75, \Psi^2 = 3, \Psi^3 = 0.75$.

- The High Dimensional Linear (*HL*) dataset is an extension of the work by Doutreligne & Varoquaux 2025 to high dimensional settings. It consists of $d = 50, 100$ covariates, 10 of which are linked to the nuisance functions and CATE by a linear function. The loading of each important covariate is drawn from a Rademacher distribution.

- The High Dimensional Polynomial (*HP*) dataset is similar to *HL* but uses a linear combination of polynomial features of degree three, including interaction terms.

- The Infant Health and Development Program (*IHDP*). The dataset consists 747 subjects with 25 real covariates, including 6 continuous and 19 binary variables, along with a simulated outcome that is both non-linear and noisy (Shalit et al., 2017).

**Variable importance for low dimensional datasets** We first compared both approaches using the previously proposed *LD* benchmark (Hines et al., 2022). Out of six covariates, the first three were important for the CATE prediction. To estimate the CATE, we used a DR-learner (Kennedy, 2023) with regularized linear models for nuisances functions and the final regression step. For PermuCATE, we used the same regularized linear model for covariate prediction and used 50 permutations. The importance of variables was estimated using a nested cross-fitting scheme. In each split split, $20\%$ of the data was left out for the importance estimation. The remaining $80\%$ was used to fit the DR-learner using the cross-validation scheme presented in the work from Kennedy (2023). We used a five-fold cross-fitting strategy for both loops. Additionally, for each sample size, we repeated these computations on 100 different random samples.

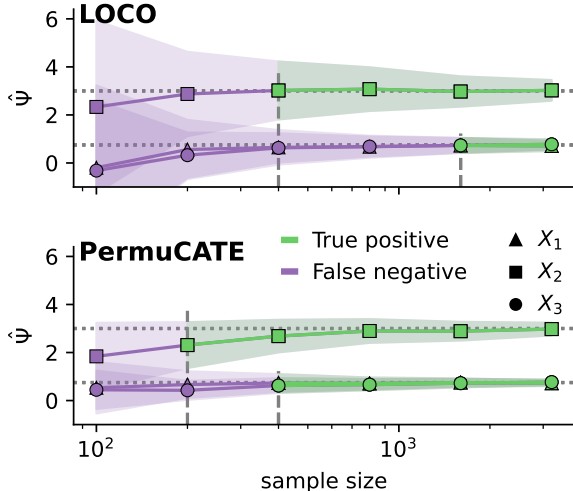

*Figure 1.* `PermuCATE` **detects important variables with greater statistical power.** Using the data-generating process proposed by (Hines et al., 2022), we compared the estimates of variable importance using `LOCO` (top subplot), the baseline proposed by the authors, and `PermuCATE`. By computing p-values, we observed at different sample sizes whether each of the three important variables was correctly identified (true positive) or missed (false negative) at a significance level $\alpha = 0.05$. For all three important variables, the minimum sample size (vertical dashed line) required to detect their importance was smaller with the `PermuCATE` approach. Both methods converged to the theoretical importance score (horizontal dotted line). The shaded areas around the curves represent Monte Carlo estimates of the standard deviation over 100 random samples.

To compute $p$-values over the different folds we used the Nadeau-Bengio corrected t-test (Nadeau & Bengio, 1999). True positives and false negative were defined with respect to a significance level of $\alpha = 0.05$. Both methods provided a false positive rate below $0.05$. As presented in Figure 1, both methods asymptotically converged to the analytical importance score represented by dotted lines. However, it can be seen that for `PermuCATE`, the standard deviation was smaller (represented by shaded areas). This larger variance of the `LOCO` estimator translated into decreased statistical power, as revealed by the minimum sample size needed to identify each of the truly important variables represented by vertical dashed lines. The following experiment aimed at studying its impact in more complex and high-dimensional settings.

**Variable importance in high dimensional settings** To study the scaling behavior of both methods, we evaluated their capacity to identify important variables in datasets with higher dimension. We used two simulation scenarios, *HL* and *HP*, which respectively used linear and polynomial link functions between the covariates and the different nui-

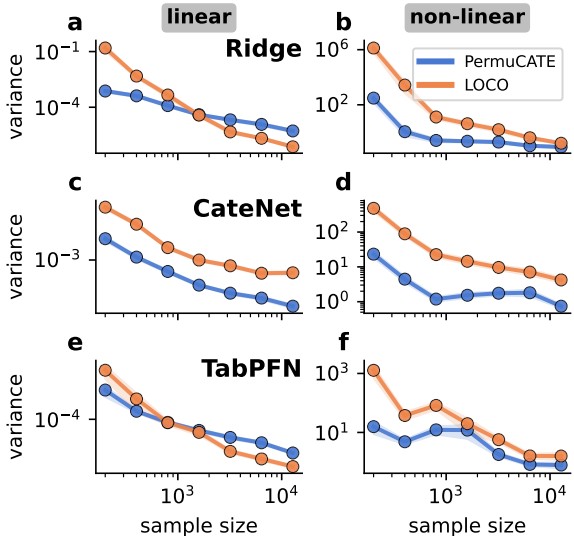

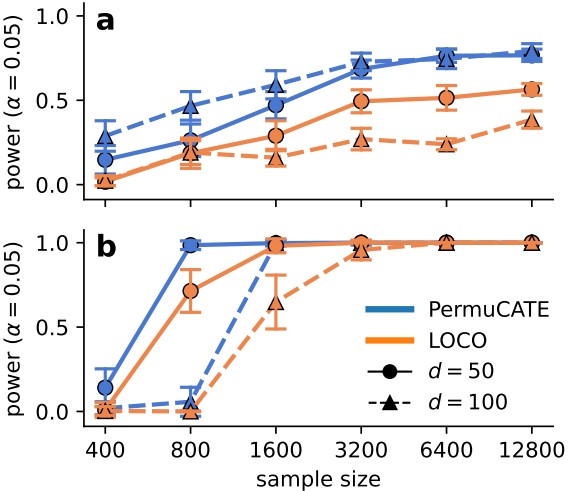

*Figure 2.* **PermuCATE demonstrates lower variance for misspecified models and small sample sizes.** The figure compares the variance of the CATE estimations when the true CATE is a linear function of the covariates (left column) and a polynomial function of degree three with interaction terms (right column). Three different CATE estimators were used: (**a, b**) DR-learner with Ridge to estimate the nuisances and pseudo-outcomes; (**c, d**) CateNet from Curth & Schaar 2021; and (**e, f**) DR-learner using TabPFN. Both simulation scenarios used (d=50) correlated Gaussian covariates with a support size of 10. Variance was assessed for one predictive variable across 100 random simulations. Error bars represent bootstrap estimates of the variance.

*Figure 3.* **The `PermuCATE` method identified more important variables in high-dimensional, linear, and complex scenarios.** (**a**) Statistical power for detecting important variables as a function of sample size on the *HP* dataset (non-linear scenario) with `PermuCATE` and `LOCO` methods. The CATE was estimated with a DR-learner using super-learners stacking gradient boosting trees and regularized linear models to estimate the nuisance functions. Results are shown for two scenarios with $d = 50$ and $d = 100$ covariates, respectively. In both cases, only 10 covariates were important. Error bars represent Monte Carlo estimates of the standard deviation over 10 simulations. (**b**) Same as (a) but using the *HL* data-generating scenario and only regularized linear models to estimate the nuisance functions.

sance functions and CATE. Both scenarios used $d = 50$ Gaussian covariates with a correlation of $0.5$. The support size was sparse using only 10 important variables. For this more complex setting, CATE was estimated using more expressive models: *CateNet*, a DL architecture proposed by Curth & Schaar 2021, *TabPFN*, a DR-learner using the pretrained transformer architecture from Hollmann et al. 2025 to estimate the nuisances and pseudo-outcomes and finally a baseline, *Ridge* as in the previous experiment. Figure 2 presents the variance when estimating the importance of a predictive covariate using `PermuCATE` and `LOCO` for different CATE estimators in linear and non-linear scenarios. In the linear setting, where the error in estimating the CATE was similar to the error in estimating the conditional distribution, the variances were comparable, with smaller values in small sample sizes for `PermuCATE` and a faster asymptotic decrease for `LOCO`. In addition, `PermuCATE` shows a systematic benefit for *CateNet* (panel **c**) where a stochastic optimization was used. However, in the non-linear scenario, `PermuCATE` demonstrated significantly lower variance, sometimes by orders of magnitude. This effect was particularly pronounced in small sample sizes and tends to diminish asymptotically. These findings, which suggest

a higher robustness to model specification and finite sample estimation error for `PermuCATE` are consistent with Equation (5) and Equation (7), which show that `LOCO` 's bias depends on the error in estimating the CATE, whereas `PermuCATE`'s does not.

We then compared their statistical power at the significance level $\alpha = 0.05$, defined by $tp/(tp + fn)$, were $tp$ is the number of true positive (variables that belong to the true support and have a *p*-value $\leq 0.05$), and $fn$ the number of false negatives. Figure 3 presents the statistical power on the *HP* dataset for different sample sizes. The CATE was computed using a DR-learner for which the different nuisances and presudo-outcomes are estimated using a stacking of gradient boosted trees and regularized linear model drawing inspiration from the popular Super Learner method (Laan et al., 2007) (further details in subsection A.11). The statistical power of `PermuCATE` was greater, especially in the non-linear setting (Figure 3, **a**). Besides this advantage, it is remarkable that `PermuCATE` suffered less from the increase of dimension as opposed to `LOCO` which presents a greater drop in power in the case where $d = 100$. This observation likely stems from the fact that the error difference presented in Equation (7) fluctuates more as the

dimension increases. Although not represented on that plot, both methods demonstrated the desired type-1 error control. Figure 3b shows the same as (a) but in the linear scenario (*HL*). Here again, `PermuCATE` appeared to benefit from increased statistical power, leading to a better identification of important variables in small sample sizes.

**Variable importance one the IHDP benchmark**
`PermuCATE` and `LOCO` were then compared on the established semi-synthetic IHDP benchmark (see datasets). For binary variables, sampling from the conditional distribution was achieved using the predicted probabilities of a logistic regression model. Three different learners were used to estimate the CATE: Causal Forest (CF) with 100 trees (Athey & Wager, 2019), deep neural networks (CATENets) from (Curth & Schaar, 2021) using the best model parameters reported in the publication, and a DR-learner incorporating tabular foundation models (Hollmann et al., 2025) to estimate nuisance functions and predict pseudo-outcomes. Consistent with Curth & Schaar 2021, results were computed over 100 simulations using a single train-test split. The semi-synthetic nature of the dataset allowed measuring the Precision in Estimating Causal Effects (PEHE), which is presented in Figure 4 **a**. It further allowed quantifying the ability of each variable importance method to recover the true support, for each learner, as shown in Figure 4 **b, c, d**. $p$-values were computed using variable importance estimates derived from the 100 simulations. The type-1 error corresponds to $fp/(fp + tn)$, where $fp$ stands for false positive and tn for true negative. The statistical power is computed by $tp/(tp + fn)$ where $tp$ stands for true positive and tn for true negative.

These results indicate that `PermuCATE` provided better estimates of variable importance under model misspecification conditions, as evidenced by its performance on the IHDP benchmark. As shown in Figure 4 a, the learner based on TabFPN outperformed the popular CATENet, itself better than the Causal Forest in terms of PEHE for estimating the CATE. For all three models, `PermuCATE` achieved significantly better AUC and statistical power for identifying important variables. This suggests that `PermuCATE` performs faovrably when applied to complex models to approximate an unknown generative mechanism. One would typically select the best-performing model to measure variable importance, as it is more likely to capture the data-generative model. However, subpanel **b** revealed that the AUC when using `LOCO` combined with a CATENet was lower than when using the CF despite the former achieving a better PEHE. This effect is due to the highly stochastic nature of the optimization process for CATENets, as it relied on the Adam algorithm, which introduces additional variance. When using deep learning models, the successive per-covariate refitting operations not only come with a finite

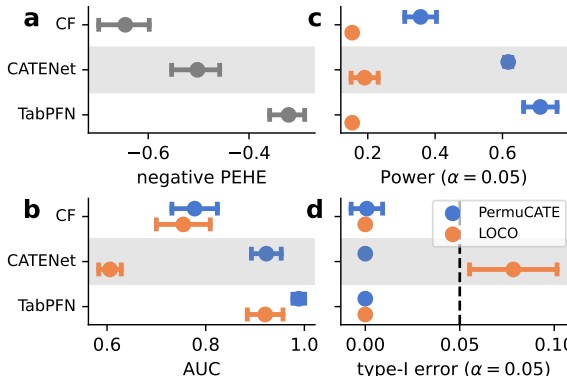

*Figure 4.* **Comparison of variable importance methods with three different learners on the IHDP benchmark.** The CATE was estimated using a Causal Forest (CF, Athey & Wager 2019), a deep neural network (CATENet, Curth & Schaar 2021), and a pre-trained tabular foundation model (TabPFN, Hollmann et al. 2025). For each learner, variable importance was estimated with `LOCO` and `PermuCATE`. **(a)** Displays the negative Precision in Estimating Heterogeneous Effects (PEHE) across the models. **(b)** presents the AUC for estimating variable importance, which can be framed as a classification task. The predicted importance is represented by 1 - $p$-value, which is subsequently utilized to compute statistical power **(c)** and the type-1 error **(d)** at the significance level alpha=0.05.

sample estimation error but also with optimization noise which is inherent to this learning paradigm. Finally, sub-panel **d** shows that both methods effectively controlled the type-1 error, except for `LOCO` when CATENet is used, likely due the aforementioned optimization noise. The experiment suggests that on real-world covariates, `PermuCATE` can achieve greater statistical power and better importance ranking than its `LOCO` counterpart, while allowing the use of state-of-the-art deep learning and foundation models.

**Complexity of the conditional distribution estimation**
The experiment using the IHDP benchmark involves real-world covariates, for which the ground truth of the conditional distribution is unknown. We found that `PermuCATE` outperformed `LOCO` (Figure 4) in terms of statistical power and AUC for detecting important features while controlling type-1 error on this dataset. These results suggest that Assumption 3.1 is realistic. Moreover, in Appendix A.13 we systematically varied the complexity of conditional distributions by generating non-Gaussian, multimodal covariates from a latent-variable model (Figure S5). Our analysis revealed that `PermuCATE` again exhibited lower variance compared to `LOCO` (Figure S6). Differences in variance trends for both methods resembled results from the Gaussian scenario shown in Figure 2. A possible explanation may be related to the fact that increased complexity of covariate distributions not only rendered con-

ditional estimation more challenging (preferentially impacting `PermuCATE`) but also CATE estimation (preferentially impacting `LOCO`). Both methods may thus have suffered in a comparable way but for different reasons, leading to stable ranking favoring `PermuCATE` for smaller sample sizes.

## 5. Discussion

In this work, we studied two estimators of variable importance in the context of CATE estimation: the `LOCO` approach (Hines et al., 2022) and the proposed `PermuCATE` algorithm. Our study focused on the pre-asymptotic finite-sample regime, aiming to provide insights on how to select variable importance methods for real-world use cases. Through theoretical analysis and simulation studies covering a wide range of scenarios, we unveiled how noise originating from the finite sample model fitting impacts both algorithms in specific ways through their respective variance terms. We wish to emphasize that this consequential aspect was neglected by most previous work analysing variable importance estimation, which focused more on type-1 error whereas our findings concern type-2 error. Our results also reveals how the finite-sample variance critically impacts real-world applications with small-to medium-sized datasets, characteristic of the biomedical field. For practical applications, we would, thus, recommend `PermuCATE` over `LOCO` as both methods present the same statistical guarantees for type-1 error control (as established for standard prediction problems by Chamma et al. 2023), while `PermuCATE` combines shorter computation times with higher power for detecting important variables for predicting treatment success. In clinical trials, this could determine whether informative signals are detected.

The proposed approach shares similarities with '*model-X*' knockoffs (KO) (Candes et al., 2018) as both involve modelling the conditional distribution of covariates. Both approaches are motivated by the assumption that estimating covariates is easier than estimating the quantity of interest (e.g., CATE) as discussed in Remark 3.3. A key difference is that '*model-X*' KO is designed for variable selection, whereas `PermuCATE` and `LOCO` focus on variable importance estimation. The latter provides richer information by measuring importance (total Sobol index) rather than just binary selection. Additionally, KO handles multiple testing, while our method ensures type-I error control. Furthermore, constructing KO variables requires complete exchangeability, a stricter condition than the conditional independence needed for `PermuCATE`. While not estimating variable importance, the conditional randomization test (CRT) from Candes et al. 2018 is more comparable in terms of its statistical properties to our approaches for individual conditional independence testing. However, CRT is much more computationally expensive.

The present work responded to two current trends in research & development for life science and healthcare: (1) the unmet need for interpretable ML methods for multimodal correlated biomedical data (beyond tabular data), and (2) CATE estimation with ML in interventional clinical studies, such as the discovery of predictive biomarkers. While this has shaped the perspective and scope of the present work, it is important to emphasize that the theoretical foundation of our study makes our results applicable beyond life sciences and medicine, including policy-impact, educational, and economic applications.

**Limitations and future work** Validating variable importance methods on real-world data is challenging due to the lack of known ground truth. We addressed this by using semi-synthetic data that combines real-world covariates and simulated outcomes. This leaves room for further exploratory analysis and validation in applications that could benefit from modern ML techniques to extract new insights from rich biomedical datasets. For example, it can be applied to prevention by examining how lifestyle and habits influence genetic factors (considered as treatments) on various outcomes like quality of life and survival. In clinical trial data, this method can support the identification of safety biomarkers to reduce the risk of adverse events and biomarkers to predict treatment response.

## Impact Statement

This paper introduces a model inspection method aimed at enhancing our understanding of heterogeneity in treatment responses in interventional studies including randomized-controlled trials but also in observational contexts. By providing robust explanations for potentially complex models, this approach can facilitate the adoption of cutting-edge ML advancements in fields where interpretability is crucial. Notably, this includes medical applications where the increasing predictive power of deep learning models, combined with the growing availability of data, holds significant promise for accelerating treatment development and supporting physician decision-making.

**Acknowledgments:** J.P., V.K, and D.E. are full-time employees of F. Hoffmann-La Roche Ltd. The employer of these authors had no influence on the design of this study nor on the interpretation of its findings. This research has received funding from the H2020 Research Infrastructures Grant EBRAIN-Health 101058516 and the VITE ANR-23-CE23-0016 and PEPR Santé numérique, Brain health Trajectories ANR-22-PESN-0012 projects.

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

# A. Supplementary Materials

## A.1. Notations

| | |
|---|---|
| $X$ | Covariates. The matrix is of dimension $n \times d$ with $n$ the sample size and $d$ the number of covariates |
| $A$ | Treatment assignment |
| $Y$ | Outcome |
| $\mu_0$ | Control response |
| $\mu_1$ | Treatment response |
| $\tau$ | Conditional Average Treatment Effect |
| $\pi$ | Propensity score |
| $\hat{f}$ | Finite sample estimate of a function $f$ |
| $X^{-j}$ | Set of covariates excluding covariate $j$ |
| $\hat{\nu}_j$ | $= \widehat{\mathbb{E}}_n[X^j \vert X^{-j}]$, Finite sample estimate of covariate $j$ from other |
| $r_j$ | $= X_j - \hat{\nu}_j$, Residual of covariate $j$ that cannot be predicted from the others |
| $\widetilde{r}_j$ | $= \text{shuffle}(r_j)$, Permuted version of the residual |
| $\Delta\hat{\beta}_{-j}$ | $d-1$ dimensional vector containing as entries, for $\{k\}_{k \neq j}$, $\left[\Delta\hat{\beta}_{-j}\right]_k = \widehat{\beta_k} - \widetilde{\beta_k}$ where $\widehat{\beta}$ are the coefficients of the model fitted on the full set of covariates and $\widetilde{\beta}$ are the coefficients of the model fitted on the subset of covariates $X^{-j}$ |
| $\Psi_j^*$ | $\mathbb{E}\left[\mathbb{V}(\tau(X))\vert X^{(-j)}\right]$ Total Sobol index, quantity of interest estimated by both `LOCO` and `PermuCATE`. |

*Table 1.*

## A.2. R- and pseudo-outcome-risk decomposition

In a scenario where the outcome $y$ is not deterministic: $y = m(x) + (a - \pi(x))\tau(x) + \epsilon(a, x)$, given a CATE estimate $\hat{\tau}$, the expectation of the pseudo-outcome risk can be formulated as,

$$
\begin{aligned}
\mathbb{E}[R_{PO}(\hat{\tau}, X, A, Y)] &= \mathbb{E}\left[(\varphi(Z) - \hat{\tau}(X))^2\right] \\
&= \mathbb{E}\left[(\frac{(Y - \mu(A, X))(A - \pi(X))}{\pi(X)(1 - \pi(X))} + \mu(1, X) - \mu(0, X) - \hat{\tau}(X))^2\right] \\
&= \mathbb{E}\left[\left(\frac{(Y - \mu(A, X))(A - \pi(X))}{\pi(X)(1 - \pi(X))} + \tau(X) - \hat{\tau}(X)\right)^2\right] \\
&= \underbrace{\mathbb{E}\left[(\tau(X) - \hat{\tau}(X))^2\right]}_{\tau-\text{risk}} + \underbrace{\mathbb{E}\left[\left(\frac{\varepsilon(X, A)(A - \pi(X))}{\pi(X)(1 - \pi(X))}\right)^2\right]}_{\text{rescaled noise term}} + \underbrace{\mathbb{E}\left[2\frac{(\tau - \hat{\tau}(X))\varepsilon(A, X)(A - \pi(X))}{\pi(X)(1 - \pi(X))}\right]}_{=0, \text{ since } \mathbb{E}[\varepsilon(A,X)]=0} \\
&= \mathbb{E}\left[(\tau(X) - \hat{\tau}(X))^2\right] + \mathbb{E}\left[\left(\frac{\varepsilon(X, A)(A - \pi(X))}{\pi(X)(1 - \pi(X))}\right)^2\right] \qquad \text{(S1)}
\end{aligned}
$$

Similarly, as shown in Doutreligne & Varoquaux 2025, , the R-risk $R_R(\hat{\tau}, X, A, Y)$ can be re-written as,

$$
\begin{aligned}
\mathbb{E}[R_R(\hat{\tau}, X, A, Y)] &= \mathbb{E}\left[((Y - m(X)) - (A - \pi(X))\hat{\tau}(X))^2\right] \\
&= \mathbb{E}\left[((A - \pi(X))\tau(X) + \varepsilon(A, X) - (A - \pi(X))\hat{\tau}(X))^2\right] \\
&= \underbrace{\mathbb{E}\left[((A - \pi(X))(\tau - \hat{\tau}(X)))^2\right]}_{\text{reweighted } \tau\text{-risk}} + \underbrace{\mathbb{E}\left[2(A - \pi(X))(\tau - \hat{\tau}(X))\varepsilon(A, X)\right]}_{=0, \text{ since } \mathbb{E}[\varepsilon(A,X)]=0} + \underbrace{\mathbb{E}\left[\varepsilon(A, X)^2\right]}_{\text{noise variance}} \\
&= \mathbb{E}\left[((1 - \pi(X))\pi(X)(\tau - \hat{\tau}(X)))^2\right] + \mathbb{E}\left[\varepsilon(A, X)^2\right] \qquad \text{(S2)}
\end{aligned}
$$

Both risks are therefore comparable in the sense that they are feasible when estimated values are used for the quantities $\hat{\pi}, \hat{\mu}_a$ and that they involve a combination of a $\tau$-MSE and a noise term (termed Bayes error in Doutreligne & Varoquaux 2025). However, the presence of the term $\pi(X)(1 - \pi(X)$ at the denominator in S1 makes the quantity less stable and thus desirable for CATE estimator selection. For instance, in a scenario where the propensity score is poorly fitted and can take extreme values (close to 0 or 1) a DR-learner might still provide reliable estimation of the CATE. However, the term $\mathbb{E}[(\frac{\varepsilon(X,A)}{\pi(X)(1-\pi(X))})^2]$ will likely take extreme values and lead to inconsistencies between the $R_{PO}$-risk and the oracle $\tau$-MSE. While this consideration holds for model selection of CATE estimators and is consistent with the findings from Doutreligne & Varoquaux 2025, as shown in S1 we did not notice any difference for the estimation of variable importance. It is likely that noise terms cancel when taking the difference in the variable importance for either LOCO or CPI. The following figure depicts the comparable experimental results up to a scaling factor for the scenario *LD*

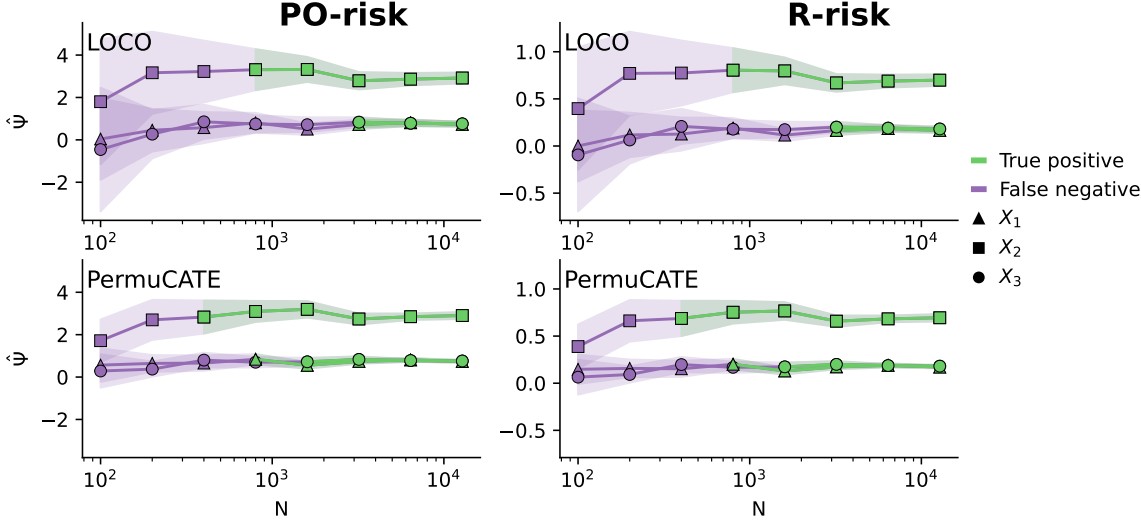

*Figure S1.* The R- and PO-risk provide, up to a scaling factor, the same measures of variable importance. This plot reproduces the results from Figure 1 (left panel) which depicts the importance measured for the 3 truly important variables $(X_1, X_2, X_3)$ in the simulation *LD* from Hines et al. 2022. In addition, the same experiment was performed using the R-risk (right panel).

### A.3. Proof of Proposition 3.2

*Proof.* The causal risk considered in this proof is the pseudo-outcome risk described in subsection A.2. Here again, $X_{P,j}$ denotes the features matrix with the $j^{th}$ feature sampled from the conditional distribution $[X_{P,j}]_j \sim X^j|X^{(-j)}$

$$2\widehat{\Psi}^j_{CPI} = \frac{1}{n_{\text{test}}} \sum_{n_{\text{test}}} \left( R(\hat{\tau}, \widehat{X}_{P,j}, A, Y) - R(\hat{\tau}, X, A, Y) \right)$$

$$= \frac{1}{n_{\text{test}}} \sum_{n_{\text{test}}} \left[ \left( \tau(X) - \hat{\tau}(\widehat{X}_{P,j}) \right)^2 \right] + \frac{1}{n_{\text{test}}} \sum_{n_{\text{test}}} \left[ \left( \frac{\varepsilon(X,A)}{\pi(X)(1-\pi(X))} \right)^2 \right]$$

$$- \frac{1}{n_{\text{test}}} \sum_{n_{\text{test}}} \left[ (\tau(X) - \hat{\tau}(X))^2 \right] - \frac{1}{n_{\text{test}}} \sum_{n_{\text{test}}} \left[ \left( \frac{\varepsilon(X,A)}{\pi(X)(1-\pi(X))} \right)^2 \right]$$

$$= \frac{1}{n_{\text{test}}} \sum_{n_{\text{test}}} \left[ \left( \tau(X) - \hat{\tau}(\widehat{X}_{P,j}) \right)^2 \right] - \frac{1}{n_{\text{test}}} \sum_{n_{\text{test}}} \left[ (\tau(X) - \hat{\tau}(X))^2 \right]$$

Then the law of large numbers provides,

$$2\widehat{\Psi}^j_{CPI} \xrightarrow{n_{\text{test}} \to \infty} \mathbb{E}[(\tau(X) - \hat{\tau}(\widehat{X}_{P,j}))^2] - \mathbb{E}[(\tau(X) - \hat{\tau}(X))^2]$$

Then using the consistency hypothesis for $\hat{\tau}$, ensures that the second error term $\mathbb{E}[(\tau(X) - \hat{\tau}(X))^2] \xrightarrow{n_{\text{train}} \to \infty} 0$ vanishes asymptotically as $\hat{\tau}$ converges to the true CATE.

Finally, thanks to the consistency hypothesis on $\hat{\nu}_j$ and Assumption 3.1 we can apply Proposition 3.3 from Reyero Lobo et al. (2025) which implies that the conditional distribution $p(X^j|X^{(-j)})$ is approximated without error, meaning that $[X_{P,j}]_j - \nu_j(X^{(-j)}) \overset{\text{i.i.d.}}{\sim} \epsilon_j$ with $\epsilon_j \perp\!\!\!\perp X^{(-j)}$. The tower property then provides

$$\mathbb{E}\left[(\tau(X) - \tau(X_{P,j}))^2\right] = \mathbb{E}\left[\mathbb{E}\left[(\tau(X) - \tau(X_{P,j}))^2 \,|X^{(-j)}\right]\right].$$

Using that $X_{P,j} \overset{\text{iid}}{\sim} X|X^{(-j)}$, we finally obtain

$$\mathbb{E}\left[\mathbb{E}\left[(\tau(X) - \tau(X_{P,j}))^2 \,|X^{(-j)}\right]\right] = \mathbb{E}\left[2\mathbb{E}\left[\left(\tau(X) - \mathbb{E}\left[\tau(X)|X^{(-j)}\right]\right)^2 |X^{(-j)}\right]\right]$$
$$= 2\mathbb{E}\left[\mathbb{V}\left[\tau(X)|X^{(-j)}\right]\right]$$

$\square$

## A.4. Proof of Proposition 3.4

*Proof.* Expectations are taken over the test set, conditionally to the training set $\mathcal{D}_{train}$. The causal risk considered in this proof is the pseudo-outcome risk described in subsection A.2. The conditional notation is omitted for readability. Here again, $X_{P,j}$ denotes the features matrix with the $j^{th}$ feature sampled from the conditional distribution $[X_{P,j}]_j \sim p(X^j|X^{(-j)})$. Additionally, we denote $\widehat{X}_{P,j}$ it's estimate using the estimator $\hat{\nu}_j$ for the conditional sampling step.

Similarly to Appendix A.3,

$$\mathbb{E}[\Psi^j_{CPI}|\mathcal{D}_{\text{train}}] = \frac{1}{2}\left(\mathbb{E}[R(\tau, X, A, Y)|\mathcal{D}_{\text{train}}] - \mathbb{E}[R(\tau, X_{P,j}, A, Y)|\mathcal{D}_{\text{train}}]\right)$$
$$= \mathbb{E}\left[\left(\tau(X) - \hat{\tau}(\widehat{X}_{P,j})\right)^2 \Big|\mathcal{D}_{\text{train}}\right] - \mathbb{E}\left[(\tau(X) - \hat{\tau}(X))^2 \Big|\mathcal{D}_{\text{train}}\right]$$

Developing the first term gives

$$\mathbb{E}\left[\left(\tau(X) - \widehat{\tau}(\widehat{X}_{P,j})\right)^2 \Big|\mathcal{D}_{\text{train}}\right] = \underbrace{\mathbb{E}\left[(\tau(X) - \tau(X_{P,j}))^2\right]}_{(A)} + \underbrace{\mathbb{E}\left[\left(\tau(X_{P,j}) - \widehat{\tau}(\widehat{X}_{P,j})\right)^2 \Big|\mathcal{D}_{\text{train}}\right]}_{(B)}$$
$$+ \underbrace{2\mathbb{E}\left[(\tau(X) - \tau(X_{P,j}))(\tau(X_{P,j}) - \widehat{\tau}(\widehat{X}_{P,j}))\Big|\mathcal{D}_{\text{train}}\right]}_{(C)}$$

The crossed-term (C) involves the estimation bias of $\tau$ and $\nu_{-j}$. This errors can be easily decomposed as

$$\mathbb{E}\left[(\tau(X) - \tau(X_{P,j}))(\tau(X_{P,j}) - \widehat{\tau}(\widehat{X}_{P,j}))\Big|\mathcal{D}_{\text{train}}\right] = \mathbb{E}\left[(\tau(X) - \tau(X_{P,j}))(\tau(X_{P,j}) - \widehat{\tau}(X_{P,j}))|\mathcal{D}_{\text{train}}\right]$$
$$+ \mathbb{E}\left[(\tau(X) - \tau(X_{P,j}))(\widehat{\tau}(X_{P,j}) - \widehat{\tau}(\widehat{X}_{P,j}))\Big|\mathcal{D}_{\text{train}}\right],$$

so we first have the error of $\tau$ and then of $\nu_j$.

Using the proof of Proposition 3.2 (A) is exactly $2\mathbb{E}\left[\mathbb{V}(\tau(X))|X^{(-j)}\right] = 2\Psi_j^*$, twice the total Sobol index. Also, (B) can be decomposed in,

$$\mathbb{E}\left[\left(\tau(X_{P,j}) - \widehat{\tau}(\widehat{X}_{P,j})\right)^2 \Big|\mathcal{D}_{\text{train}}\right] = \mathbb{E}\left[(\tau(X_{P,j}) - \widehat{\tau}(X_{P,j}))^2|\mathcal{D}_{\text{train}}\right] + \mathbb{E}\left[(\widehat{\tau}(X_{P,j}) - \widehat{\tau}(\widehat{X}_{P,j}))^2 \Big|\mathcal{D}_{\text{train}}\right]$$
$$+ 2\mathbb{E}\left[(\tau(X_{P,j}) - \widehat{\tau}(X_{P,j}))(\widehat{\tau}(X_{P,j}) - \widehat{\tau}(\widehat{X}_{P,j}))\Big|\mathcal{D}_{\text{train}}\right].$$

Then, using that $X_{P,j} \overset{\text{i.i.d.}}{\sim} X$, we have that $\mathbb{E}\left[(\tau(X_{P,j}) - \widehat{\tau}(X_{P,j}))^2 \middle| \mathcal{D}_{\text{train}}\right] = \mathbb{E}\left[(\tau(X) - \widehat{\tau}(X))^2 \middle| \mathcal{D}_{\text{train}}\right]$. Therefore, the error in estimating $\tau$ cancels out for PermuCATE, in contrast to LOCO, which depends on this error. Finally the cross term being a product of zero-mean errors coming from independent optimizations can be neglected, leading to,

$$\mathbb{E}[\widehat{\Psi}^j_{CPI}|\mathcal{D}_{\text{train}}] = 2\Psi^*_j + \mathbb{E}\left[(\widehat{\tau}(X_{P,j}) - \widehat{\tau}(\widehat{X}_{P,j}))^2 \middle| \mathcal{D}_{\text{train}}\right] \tag{9}$$

Lastly $\widehat{\tau}$ being K-Lipschitz, we get

$$\mathbb{E}\left[\left(\widehat{\tau}(X_{P,j}) - \widehat{\tau}(\widehat{X}_{P,j})\right)^2 \middle| \mathcal{D}_{\text{train}}\right] \leq \mathbb{E}\left[K\left|\left|X_{P,j} - \widehat{X}_{P,j}\right|\right|^2 \middle| \mathcal{D}_{\text{train}}\right]$$

Then, recalling that the $j^{th}$ covariate of $X_{P,j}$ is obtained by sampling from the conditional distribution, it corresponds to $\nu_j(X^{(-j)}) + (X'^j - \nu_j(X'^{(-j)}))$, were $X' \perp\!\!\!\perp X$ and $X^j$ is its $j^{th}$ covariate. It therefore comes that

$$\mathbb{E}\left[\left(\widehat{\tau}(X_{P,j}) - \widehat{\tau}(\widehat{X}_{P,j})\right)^2 \middle| \mathcal{D}_{\text{train}}\right] \leq K\mathbb{E}\left[(\nu_j(X^{(-j)}) - \widehat{\nu}_j(X^{(-j)}) + \widehat{\nu}_j(X'^{(-j)}) - \nu_j(X'^{(-j)}))^2 \middle| \mathcal{D}_{\text{train}}\right]$$

$$\leq 2K\,\mathbb{E}\left[(\nu_j(X^{(-j)}) - \widehat{\nu}_j(X^{(-j)}))^2 \middle| \mathcal{D}_{\text{train}}\right],$$

where we have used that $X \overset{\text{i.i.d.}}{\sim} X'$. Finally, denoting $||\nu_j - \hat{\nu}_j||^2 = \mathbb{E}\left[||\nu_j - \widehat{\nu}_j||^2 \middle| \mathcal{D}_{\text{train}}\right]$, we obtain,

$$\mathbb{E}\left[\widehat{\Psi}^j_{\text{CPI}} \middle| \mathcal{D}_{\text{train}}\right] = \Psi^*_j + O_P(||\nu_j - \widehat{\nu}_j||^2),$$

where the $O_P$ notation denotes bounded in probability. $\qquad\square$

## A.5. Proof of Proposition 3.5

*Proof.*

$$\mathbb{E}\left[\widehat{\Psi}^j_{\text{LOCO}} \middle| \mathcal{D}_{\text{train}}\right] = \mathbb{E}\left[R\left(\hat{\tau}^{-j}, X^{-j}, A, Y\right) - R\left(\hat{\tau}, X, A, Y\right)\middle| \mathcal{D}_{\text{train}}\right]$$

$$= \mathbb{E}\left[\left(\tau - \widehat{\tau}_{-j}(X^{-j})\right)^2 - (\tau - \widehat{\tau}(X))^2 \middle| \mathcal{D}_{\text{train}}\right].$$

On the one hand, we have that

$$\mathbb{E}\left[\left(\tau - \widehat{\tau}_{-j}(X^{-j})\right)^2 \middle| \mathcal{D}_{\text{train}}\right] = \mathbb{E}\left[\left(\tau(X) - \widehat{\tau}_{-j}(X^{-j})\right)^2 \middle| \mathcal{D}_{\text{train}}\right]$$

$$= \mathbb{E}\left[\left(\tau_{-j}(X^{-j}) - \widehat{\tau}_{-j}(X^{-j})\right)^2 \middle| \mathcal{D}_{\text{train}}\right] + \mathbb{E}\left[\left(\tau(X) - \tau_{-j}(X^{-j})\right)^2 \middle| \mathcal{D}_{\text{train}}\right],$$

where we have used that

$$\mathbb{E}\left[(\tau_{-j}(X^{-j}) - \widehat{\tau}_{-j}(X^{-j}))(\tau(X) - \tau_{-j}(X^{-j}))|\mathcal{D}_{\text{train}}\right]$$
$$= \mathbb{E}\left[(\tau_{-j}(X^{-j}) - \widehat{\tau}_{-j}(X^{-j}))\mathbb{E}\left[(\tau(X) - \tau_{-j}(X^{-j}))|X^{-j}, \mathcal{D}_{\text{train}}\right]\middle|\mathcal{D}_{\text{train}}\right]$$
$$= 0.$$

We note that $\mathbb{E}\left[\left(\tau(X) - \tau_{-j}(X^{-j})\right)^2\right]$ is exactly $\mathbb{E}\left[\mathbb{V}(\tau(X))|X^{(-j)}\right] = \Psi^*_j$, the total Sobol index. Combining both terms, completes the proof,

$$\mathbb{E}\left[\widehat{\Psi}^j_{\text{LOCO}} \middle| \mathcal{D}_{\text{train}}\right] = \Psi^*_j + \mathbb{E}\left[\left(\tau_{-j}(X^{-j}) - \widehat{\tau}_{-j}(X^{-j})\right)^2 \middle| \mathcal{D}_{\text{train}}\right] - \mathbb{E}\left[(\tau(X) - \widehat{\tau}(X))^2 \middle| \mathcal{D}_{\text{train}}\right] \tag{10}$$

$$= \Psi^*_j + O_P(||\tau_{-j} - \widehat{\tau}_{-j}||^2 - ||\tau - \widehat{\tau}||^2)$$

Were $||\cdot||^2$ is the L2-norm with the fixed train set $\mathcal{D}_{\text{train}}$ $\qquad\square$

### A.6. Finite sample variance of `LOCO` and `CPI` in a linear scenario

This subsection presents the analytical variance of the `CPI` and `LOCO` estimators in the linear setting it builds on top of results from Reyero Lobo et al. 2025. In such a scenario, we assume the cate to be linear $\tau(X) = X\beta$ and we consider linear estimators $\widehat{\tau}(X) = X\widehat{\beta}$.

### A.7. LOCO

For `LOCO`, recalling Equation (10)

$$\mathbb{E}\left[\widehat{\Psi}_{\text{LOCO}}^{j}\Big|\mathcal{D}_{\text{train}}\right] = \Psi_j^* + \mathbb{E}\left[\left(\tau_{-j}(X^{-j}) - \widehat{\tau}_{-j}(X^{-j})\right)^2\Big|\mathcal{D}_{\text{train}}\right] - \mathbb{E}\left[(\tau(X) - \widehat{\tau}(X))^2\big|\mathcal{D}_{\text{train}}\right]$$

Considering linear models, we get:

$$\mathbb{E}\left[\left(\tau_{-j}(X^{-j}) - \widehat{\tau}_{-j}(X^{-j})\right)^2\Big|\mathcal{D}_{\text{train}}\right] = \mathbb{E}\left[\left(X^{-j}\beta^{(-j)} - X^{-j}\widehat{\beta}^{(-j)}\right)^2\Big|\mathcal{D}_{\text{train}}\right]$$

Denoting $\Delta\hat{\beta}_{(-j)} = \beta^{(-j)} - \widehat{\beta}^{(-j)}$ we get,

$$\mathbb{E}\left[\left(\tau_{-j}(X^{-j}) - \widehat{\tau}_{-j}(X^{-j})\right)^2\Big|\mathcal{D}_{\text{train}}\right] = ||\Delta\hat{\beta}_{(-j)}||^2_{\Sigma_{-j}}$$

with $\Sigma_{-j} = \mathbb{E}\left[X^{(-j)\top}X^{(-j)}\right]$ and $||A||_{\Sigma} = A^{\top}\Sigma A$. Using a similar reasoning leads to,

$$\mathbb{E}\left[\widehat{\Psi}_{\text{LOCO}}^{j}\Big|\mathcal{D}_{\text{train}}\right] = \Psi_j^* + ||\Delta\hat{\beta}_{(-j)}||^2_{\Sigma_{-j}} - ||\Delta\hat{\beta}||^2_{\Sigma}$$

Finally, using assuming that independent samples are used to fit $\tau$ and $\tau_{-j}$ (for instance using sample splitting similarly to Williamson et al. 2023) we get

$$\text{var}\left(\mathbb{E}\left[\widehat{\Psi}_{\text{LOCO}}^{j}\Big|\mathcal{D}_{\text{train}}\right]\right) = \text{var}\left(||\Delta\hat{\beta}_{(-j)}||^2_{\Sigma_{-j}}\right) + \text{var}\left(||\Delta\hat{\beta}||^2_{\Sigma}\right)$$

### A.8. PermuCATE

Regarding CPI, we recall from Equation (9),

$$\mathbb{E}[\widehat{\Psi}_{CPI}^{j}|\mathcal{D}_{\text{train}}] = \mathbb{E}\left[(\tau(X) - \tau(X_{P,j}))^2\right] + \mathbb{E}\left[(\widehat{\tau}(X_{P,j}) - \widehat{\tau}(\widehat{X}_{P,j}))^2\Big|\mathcal{D}_{\text{train}}\right]$$

Concerning A, we recall that the $j^{th}$ covariate of $X_{P,j}$ corresponds to $\nu_j(X^{(-j)}) + (X'^j - \nu_j(X'^{(-j)}))$ with $X' \perp\!\!\!\perp X$. That all other covariates remain unchanged and that we assume $\tau = \beta X$. Consequently,

$$\mathbb{E}\left[(\widehat{\tau}(X_{P,j}) - \widehat{\tau}(\widehat{X}_{P,j}))^2\Big|\mathcal{D}_{\text{train}}\right] = \mathbb{E}\left[\widehat{\beta}_j^2\left(\nu_j(X^{(-j)}) + (X'^j - \nu_j(X'^{(-j)})) - \widehat{\nu}_j(X^{(-j)}) - (X'^j - \widehat{\nu}_j(X'^{(-j)}))\right)^2\Big|\mathcal{D}_{\text{train}}\right]$$

$$= \widehat{\beta}_j^2\mathbb{E}\left[\left(\nu_j(X^{(-j)}) - \nu_j(X'^{(-j)}) - \widehat{\nu}_j(X^{(-j)}) + \widehat{\nu}_j(X'^{(-j)})\right)^2\Big|\mathcal{D}_{\text{train}}\right]$$

Then considering Gaussian covariates, the model $\nu$ is linear, giving $\nu_j(X^{(-j)}) = X^{(-j)}\gamma_{(-j)} + \varepsilon_j$,

$$\mathbb{E}\left[\widehat{\tau}(X_{P,j}) - \widehat{\tau}(\widehat{X}_{P,j}))^2\Big|\mathcal{D}_{\text{train}}\right] = \widehat{\beta}_j^2\mathbb{E}\left[\left(\Delta\widehat{\gamma}(X^{(-j)} - X'^{(-j)})\right)^2\Big|\mathcal{D}_{\text{train}}\right]$$

$$= 2\widehat{\beta}_j^2||\Delta\widehat{\gamma}||^2_{\Sigma_{-j}}$$

where $\Delta\widehat{\gamma} = \gamma - \widehat{\gamma}$, is the coefficient difference when estimating the linear function $\nu_j$ and $\Sigma_{-j} = \mathbb{E}\left[X^{(-j)\top}X\right]$. To obtain the last line, we used that $X' \perp\!\!\!\perp X$. Therefore when using the rescaled version $\frac{1}{2}\widehat{\Psi}_{CPI}^{j}$, we obtain,

$$\text{var}\left(\mathbb{E}[\widehat{\Psi}_{CPI}^{j}|\mathcal{D}_{\text{train}}]\right) = \text{var}(\widehat{\beta}_j^2||\Delta\widehat{\gamma}||^2_{\Sigma_{-j}})$$

## A.9. Variance comparison in a linear scenario

An empirical comparison of the variance of `LOCO` and `PermuCATE` is shown in Fig S2.

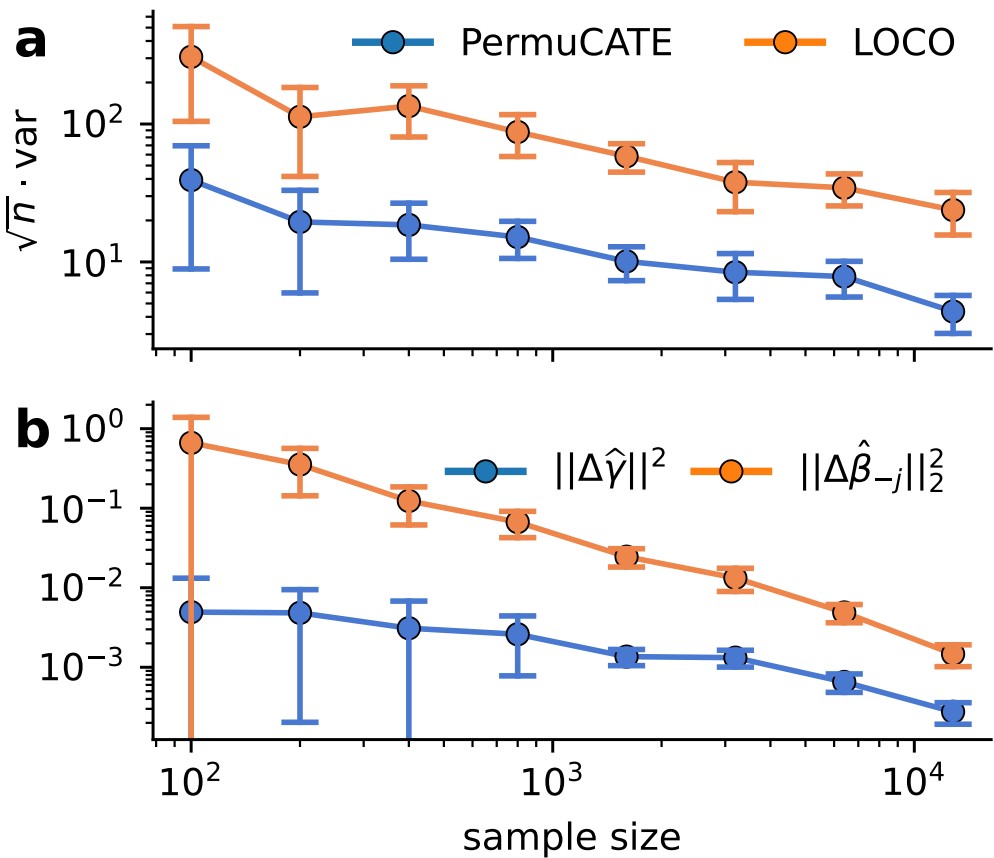

*Figure S2.* **The lower variance of `PermuCATE` is explained by finite-sample estimation error.** (a) $\sqrt{n}$ times the variance of the importance estimate of $X_2$ ($\Psi^{(2)}$) on the *LD* dataset using `LOCO` and `PermuCATE` for different sample sizes. (**b**) Dependence of different error terms on sample size. For `PermuCATE`, this noise can be attributed to the the error in the estimation of $\hat{\nu}_j = \hat{\mathbb{E}}_n[X_j|X^{-j}]$ and is captured by its $||\Delta\hat{\gamma}||^2$. For `LOCO`, it is captured by the fluctuations in the estimation of the model coefficients of the model introduced by the refitting procedure and is captured by $||\Delta\beta_{-j}||_2^2$. The correlation structure from the *LD* dataset has been removed for clarity.

## A.10. Datasets

**Low Dimensional (LD) dataset**   Taken from the work of Hines et al. 2022:

$$(X_1, X_2), (X3, X_4), (X_5, X_6) \sim \mathcal{N}(0, \begin{bmatrix} 1 & 0.5 \\ 0.5 & 1 \end{bmatrix})$$
$$\tau(X) = X_1 + 2X_2 + X_3$$
$$\pi(X) = \text{expit}\left(-0.4X_1 + 0.1X_1X_2 + 0.25X_5\right)$$
$$A \sim \text{Bernoulli}(\pi(X))$$
$$\mu_0(X) = X_3 - X_6$$
$$Y \sim \mu_0(X) + A\tau(X) + \mathcal{N}(0,3)$$

**The High Dimensional Linear (*HL*) dataset**  Create the sets $S^\pi, S^{\mu_0}, S^\tau$ of important variables, respectively for the functions $\pi, \mu_0, \tau$ by randomly drawing without replacement $d_{imp} = 10$ important variables from the set $\{1, \cdots, d\}$ of possible variables. We denote the vector of covariates $X = [x_1, \cdots, x_d]$. Covariates follow a random normal distribution and with a correlation coefficient that can be modified.

$$\pi(X) = \text{expit}\left(\sum_{i=1}^{d} x_i \beta_i^\pi\right)$$

$$\beta_i^\pi = \begin{cases} 0 & \text{if } i \notin S^\pi \\ \beta_i^\pi \sim \text{Rademacher}(0.5) & \text{if } i \in S^\pi \end{cases}$$

A Rademacher variable takes value in $\{-1, 1\}$ with equal probability.

$$\mu_0(X) = \sum_{i=1}^{d} x_i \beta_i^{\mu_0}$$

$$\beta_i^{\mu_0} = \begin{cases} 0 & \text{if } i \notin S^{\mu_0} \\ \beta_i^{\mu_0} \sim \text{Rademacher}(0.5) & \text{if } i \in S^{\mu_0} \end{cases}$$

$$\tau(X) = \sum_{i=1}^{d} x_i \beta_i^\tau$$

$$\beta_i^\tau = \begin{cases} 0 & \text{if } i \notin S^\tau \\ \beta_i^\tau \sim \text{Rademacher}(0.5) & \text{if } i \in S^\tau \end{cases}$$

We then construct,

$$A \sim \text{Bernoulli}(\pi(X))$$
$$Y = (1 - e) \cdot \mu_0(X) + e \cdot A\tau(X) + \mathcal{N}(0, 1)$$

Where $e \in [0, 1]$ denotes the effect size which controls the strength of the treatment effect.

**The High Dimensional Polynomial (*HL*) dataset**  This dataset has a similar construction as *HL* but uses polynomial features. Consequently,

$$\pi(X) = \text{expit}\left(\sum_{i=1}^{d}\sum_{j=1}^{d}\sum_{k=1}^{d} x_i x_j x_k \beta_{i,j,k}^\pi\right)$$

$$\beta_{i,j,k}^\pi = \begin{cases} 0 & \text{if } i \notin S^\pi \vee j \notin S^\pi \vee k \notin S^\pi \\ \beta_{i,j,k}^\pi \sim \text{Rademacher}(0.5) & \text{if } i \in S^\pi \wedge j \in S^\pi \wedge k \in S^\pi \end{cases}$$

$\mu_0$ and $\tau$ follow a similar construction as in *LD* using the same polynomial features.

In addition, we also control the proportion of treated units by computing the $p^{th}$ quantile of the propensity scores that we denote $Q_p$ in order to avoid extreme imbalances between the number of units in the treated and control group.

$$A \sim \text{Bernoulli}(\pi(X) - Q_p)$$
$$Y = (1 - e) \cdot \mu_0(X) + e \cdot A\tau(X) + \mathcal{N}(0, 1)$$

**Infant Health and Development Program (IHDP)**  This benchmark consists in data from a clinical trial studuying premature the effect of intervention in a population of premature infants. The Intervention Group received home visits, attendance at a special child development center, and pediatric follow-up. It contains 747 subjects and 25 covariates (6

continuous, 19 binary) describing infants and their mothers. It is imbalanced, with 139 treated and 608 controls, making the CATE estimation more challenging than in a matched clinical trial. While covariates are real, the outcomes are simulated using,

$$Y \sim A\mathcal{N}(\exp X\beta - \omega, 1) + (1 - A)\mathcal{N}(\exp(X + W)\beta, 1)$$

where $\beta$ is samples from $\{0, 0.1, 0.2, 0.3, 0.4\}$ with probabilities $\{0.6, 0.1, 0.1, 0.1, 0.1\}$. $W$ and $\omega$ are offset terms as described in the publication from Shalit et al. 2017. Similar to Shalit et al. 2017; Curth & Schaar 2021, we used 100 repetitions of the simulations.

## A.11. Hyper-parameter search

All hyper-parameters were optimized using a nested cross-validation loop. For linear models, we used the scikit-learn implementation `RidgeCV` for regression and `LogisticRegressionCV` with a range of penalization strength from $10^-3$ to $10^3$ with 10 logarithmically spaced values. For the gradient boosting tree we used the implementations `HistGradientBoostingClassifier` and `HistGradientBoostingRegressor` respectively for regression and classification. After using a randomized search for hyper-parameters we used a learning rate of $0.1$ (range explored: $[10^{-3}, 10^3]$) and a maximum number of leaves for each tree of 10 (range explored: $[10, 100]$).

## A.12. Variable importance using grouping

Although not emphasized in the main text, the `PermuCATE` approach is compatible with grouping strategies, which can be useful when dealing with highly correlated variables that lead to vanishing conditional importance. To address this, Chamma et al. 2024 proposed extending the conditional permutation importance framework to groups of variables. To empirically demonstrate this, we increased the dimensionality of the IHDP dataset by adding noisy copies of the variables—Gaussian noise was added to continuous covariates and binary labels were flipped with a 0.1 probability. Groups were formed by combining original variables with their noisy copies, and conditional importance was measured for these groups. The previous metrics were adapted accordingly: type-1 error involved identifying a group of null variables, and a true positive involved a group with at least one predictive variable. As shown in Figure S3, which is analogous to Figure 4 but with dimensionality on the y-axis, increasing dimensionality negatively impacts performance since no additional information is provided. Interestingly, doubling the dimensionality did not significantly harm power or AUC for predicting importance but did reduce the PEHE. As before, CATENet models exhibited higher type-1 error rates, especially with increasing dimensionality, likely due to their highly stochastic optimization process.

## A.13. Non-Gaussian variable distributions

In this section, we propose an additional experiment where we revisit the simulation study presented in Figure 2 also to vary the complexity of the conditional distributions. To do so we use simulations inspired by (Hollmann et al., 2025), we sample the covariates using a latent variable model. We first sample latent variables from mixtures of Gaussians and then generate observed covariates through a non-linear transformation with interaction terms of the latent variables. This results in non-Gaussian covariates with complex conditional distributions, illustrated in Figure S5, **a**) presents the marginal distributions, **b**) the kurtosis of generated variables (blue) compared with the kurtosis for Gaussians (orange) for comparison, and **c**) their correlation matrix. Similarly to Figure 2, we generate 2 CATE functions, one linear and one non-linear and compared the variance of `PermuCATE` and LOCO for three different models at varying sample sizes. The results shown in Figure S6 reveal a lower variance for `PermuCATE` compared to LOCO similar to Figure 2. Our interpretation is that the complexity of covariates distributions also affects LOCO, by making the CATE harder to estimate.

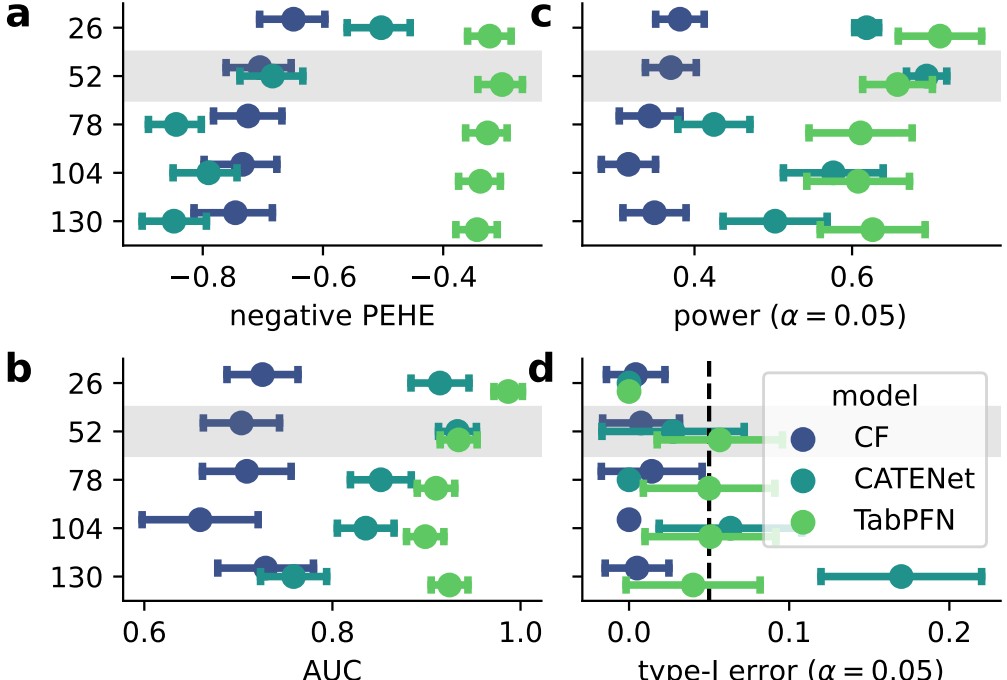

*Figure S3.* **Comparison of variable importance methods the IHDP benchmark using groups of variables.** The CATE was estimated using a Causal Forest (CF, Athey & Wager 2019), a deep neural network (CATENet, Curth & Schaar 2021). For each learner, variable importance was estimated with `PermuCATE`. The dimensionality was artificially increased by creating **(a)** Displays the negative Precision in Estimating Heterogeneous Effects (PEHE) across the models. **(b)** presents the AUC for estimating variable importance, which can be framed as a classification task. The predicted importance is represented by 1 - $p$-value, which is subsequently utilized to compute statistical power **(c)** and the type-1 error **(d)** at the significance level alpha=0.05.

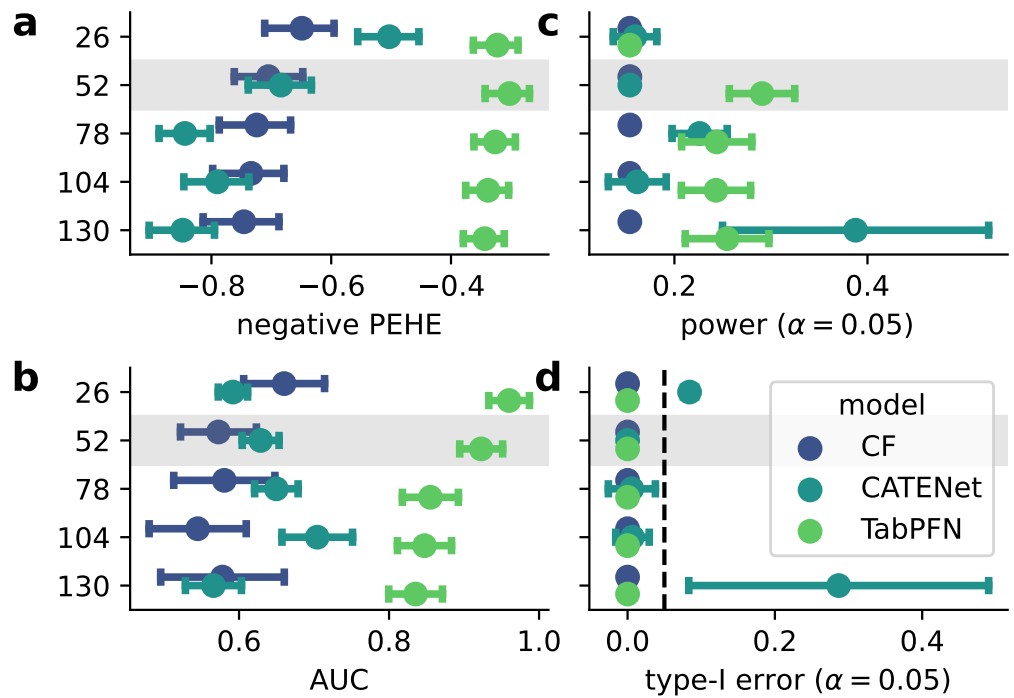

*Figure S4.* Same as Figure S3 but using Leave One Group Out (LOGO)

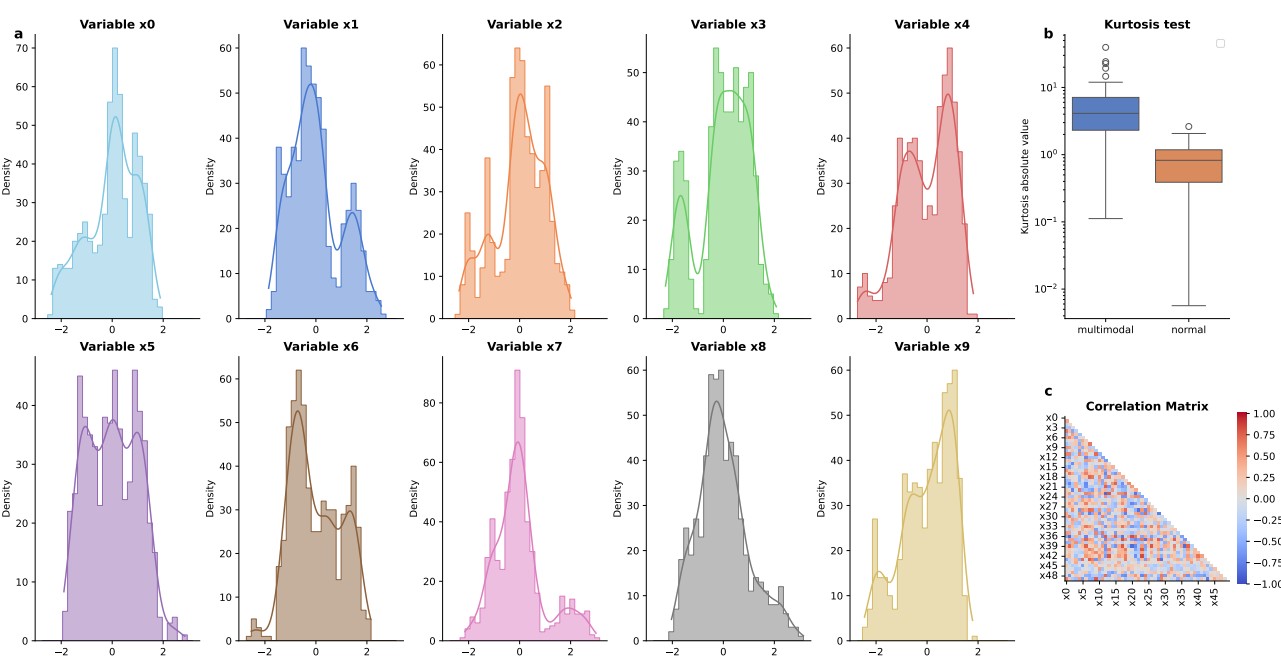

*Figure S5.* **Simulation with non-Gaussian variable distributions a**) presents the marginal distributions, **b**) the kurtosis of generated variables (blue) compared with the kurtosis for Gaussians (orange) for comparison, and **c**) their correlation matrix.

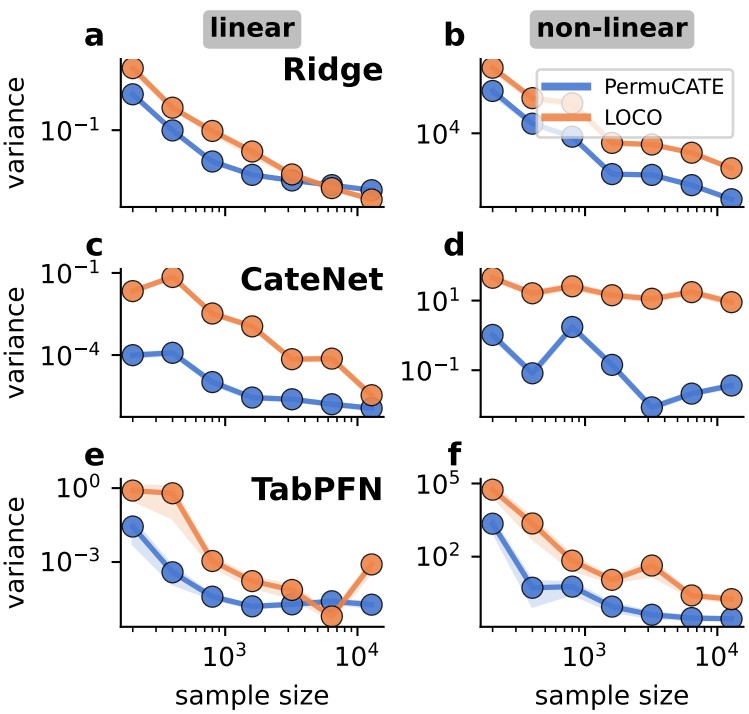

*Figure S6.* Same as Figure 2 but using the variable distributions described in Figure S5

