# OpenReview forum: "Measuring Variable Importance in Heterogeneous Treatment Effects with Confidence"
_ICML.cc/2025/Conference — ICML 2025 poster_

### Official Review · Reviewer_VGy6 · 2025-03-05

**Overall Recommendation:** 4

**Summary:**

This paper tackles measuring variable importance in conditional average treatment effect (CATE) functions. One of the few current approaches consists in applying the LOCO method to CATE estimation ; as the CPI method is an alternative to LOCO, authors propose applying CPI to CATE estimation. They prove the consistency of both CPI and LOCO to the total Sobol index for a specific risk function, where the variance of CPI is driven by its less error-prone missing covariate estimation procedure. They empirically show that CPI leads to faster convergence and lower variance than LOCO on CATE variable importance measure problems.

(EDIT : updated my score following the rebuttal)

**Claims And Evidence:**

Overall, the method seems well-justified ; I'll just have a slight concern for Assumption 3.1 that looks like the backbone of the method and its advantages and that I find lacks a bit of justification. Does it come from the past literature on CPI? How is it generally justified?

**Essential References Not Discussed:**

Not to my knowledge

**Experimental Designs Or Analyses:**

Yes, and everything looks good

**Methods And Evaluation Criteria:**

Yes

**Other Comments Or Suggestions:**

I strongly suggest you explain in greater depth the total Sobol index, eg where it comes from and why it is very appropriate for variable importance. It seems like a critical quantity in your work, as the ground-truth measure to which either CPI and LOCO is expected to converge. You might explain why it is to be taken as a ground-truth.

I do not see any typos.

**Other Strengths And Weaknesses:**

The paper is original in the sense that it is a combination of existing ideas ; it might be seen as a relatively low-hanging fruit but the theoretical and experimental evaluations done by authors increase the significance of their work. The paper is generally clear, except on the point with Assumption 3.1 above and the total Sobol index, which seems to a critical notion of the paper (see questions below)

**Questions For Authors:**

As said previously, can you please give details (literature + relevance) for :

1) Assumption 3.1

2) The total Sobol index?

**Relation To Broader Scientific Literature:**

As the LOCO method was applied to CATE estimation, and the CPI method was not, the submission fills this gap. It does justify theoretically and experimentally the use of CPI compared to LOCO.

**Theoretical Claims:**

I did quickly and did not find any issues

---

> ### Author Rebuttal · Authors · 2025-03-30
>
> We would like to thank the reviewer. The comments and questions are relevant and will help us improve the paper.
>
> # Discussion of assumption 3.1
>  - Assumption 3.1, that a given covariate can be decomposed as a function of the other covariates plus an additive independent noise term is quite common in the literature. It was especially discussed in (Candlès et al., Panning for Gold, 2017), introducing the model-x knockoff framework. In section 1.3, the authors write, "assuming we know everything about the covariate distribution but nothing about the conditional distribution $Y |X1, . . . , Xp$". We discuss the relevance of this assumption in remark 3.3 of the main paper. It is motivated by plausible scenarios where, for instance, the relationship between the covariates is much simpler than the relationship between the covariates and the outcome. For example, in genetics, the relationship between single nucleotide polymorphisms (SNPs) is easier to model than the relationship between SNPs and a disease. Also, when a large number of unsupervised data is available, learning the conditional distribution of the covariates can be easier than learning the outcome, for which only a few labeled examples are available. Finally, a temporal aspect may also be considered, where the covariates are observed simultaneously while the outcome is observed later.
>
>
>  - This assumption was also made in previous published work on CPI (Chamma et al, 2023, NeurIPS), which also revealed that in the context of prediction tasks, CPI was a robust method for variable importance estimation.
>
>
>  - We also refer the reviewer to the response provided to Reviewer jT2T, section "Complexity of the conditional distribution estimation." We clarify that the IHDP benchmark uses real-world data, where this assumption cannot be verified. And present an additional simulation experiment where the dependency structure of the covariates is made more complex, using a latent variable model. PermuCATE still shows a lower variance than LOCO in this scenario.
>
>
> # Description of the total Sobol index
>  - The total Sobol index is a well-studied quantity (see, for instance, [1, 2, 3, 4]) that measures the influence of a variable or group of variables on the output of a (possibly complex) model. For a model $\mu$ predicting an outcome $y$ given covariates $X$, the total Sobol index of a variable $j$ (or group) is given by: $$\Psi^* = \mathbb{E}[\mathbb{V}[\mu(X) | X^{-j}]] = \mathbb{E}[(y-\mu_{-j}(X^{-j}))^2]-\mathbb{E}[(y-\mu(X))^2],$$
> where $\mu_{-j}$ is the model where the variable $j$ is removed.
>
>  - By writing the total Sobol index as a loss difference, we can see that it measures the loss increase when removing the variable $j$ from the model, which provides a measure of the importance of the variable. It can also be seen as an unnormalized generalized ANOVA (difference of $R^2$) [5]:
>
> $$\Psi^* = \mathbb{V}(y)\left[\left[1-\frac{\mathbb{E}[(y-\mu(X))^2]}{\mathbb{V}(y)}\right]-\left[1-\frac{\mathbb{E}[(y-\mu_{-j}(X^{-j}))^2]}{\mathbb{V}(y)}\right]\right] .$$
>
>  - While this is true for predictive models, extending this definition to CATE estimation is not straightforward due to the fundamental problem of causal inference, the ground truth CATE is not observable (in clinical trials, patients are either in the treatment or control group, not both).
>
>  - It can be seen from the above equation that LOCO is a plug-in estimator of the total Sobol index, where the model $\mu$ is replaced its finite sample estimate $\hat{\mu}$. We show in the paper why this plug-in estimator is not the best choice for estimating the total Sobol index, especially in finite sample.
>
> We will integrate this intuitive presentation of the total Sobol index in the final version of the paper.
>
> - [1] Sobol, IM. "Global sensitivity indices for nonlinear mathematical models and their Monte Carlo estimates." Mathematics and computers in simulation, 2001
> - [2] Bénard, C. et al. Mean decrease accuracy for random forests: inconsistency, and a practical solution via the Sobol-MDA, Biometrika, 2022
> - [3] Hooker, G. et al. Unrestricted permutation forces extrapolation: variable importance requires at least one more model, or there is no free variable importance. Stat Comput, 2021
> - [4] Williamson, B. et al. A General Framework for Inference on Algorithm-Agnostic Variable Importance. Journal of the American Statistical Association, 2022
> - [5] Williamson, B et al. Nonparametric variable importance assessment using machine learning techniques. Biometrics, 2021

---

> > ### Comment · Reviewer_VGy6 · 2025-04-04
> >
> > Many thanks, this addresses my concerns. I will update my score to a clear Accept.

---

### Official Review · Reviewer_GH6j · 2025-03-17

**Overall Recommendation:** 3

**Summary:**

The submission describe a variable importance method (PermuCATE) generalizing CPI (Chamma et al. 2023),
a theoretical analysis of PermuCATE is performed showing the behaviour of the estimator in finite sample.
Extensive experiments are implemented over a variety of datasets comparing PermuCATE against LOCO.

**Claims And Evidence:**

Yes

**Essential References Not Discussed:**

I am not aware of previous references not discussed.

**Experimental Designs Or Analyses:**

yes the experimental design is appropriate and sound

**Methods And Evaluation Criteria:**

yes they make sense

**Other Comments Or Suggestions:**

no

**Other Strengths And Weaknesses:**

while the paper is well written in general I fell it should provide a better description and connection between CATE and variable importance.

For instance Verdinelli & Wasserman, 2023 make a distinction between 3 types of variables importance: population  (how important is age in a regression function?), algorithmic (how much did age affect the estimated value of the regression) and causal (ow would the outcome have
changed if Mary had been 5 years younger).
The present submission deals with the causal intepretation, being it the CATE. While  Verdinelli & Wasserman, (2023) focus on the population case. This is justified by the authors of this submission citing Hines et al., 2022 (which is a preprint not published yet?), I skimmed Hines et al., 2022 but I could not pinpoint where the connection can be made.
Meaning, how feature importance of the CATE tell us something about effect modification or differences in the causal effect? maybe this is naive but I have intuitive feeling that is not the same.

**Questions For Authors:**

For instance Verdinelli & Wasserman, 2023 make a distinction between 3 types of variables importance: population  (how important is age in a regression function?), algorithmic (how much did age affect the estimated value of the regression) and causal (ow would the outcome have
changed if Mary had been 5 years younger).
The present submission deals with the causal intepretation, being it the CATE. While  Verdinelli & Wasserman, (2023) focus on the population case. This is justified by the authors of this submission citing Hines et al., 2022 (which is a preprint not published yet?), I skimmed Hines et al., 2022 but I could not pinpoint where the connection can be made.
Meaning, how feature importance of the CATE tell us something about effect modification or differences in the causal effect? maybe this is naive but I have intuitive feeling that is not the same.

Moreover it is not completely clear to me the problem setting, especially the goal of this? the objective is to obtain causal estimates on effect modification?

**Relation To Broader Scientific Literature:**

the method is an extension of previous work (Chamma et al. 2023,  Verdinelli & Wasserman, 2023 and  Hines et al., 2022).
The basic references of this work are all preprints,

**Theoretical Claims:**

no i did not check proofs in the appendix

---

> ### Author Rebuttal · Authors · 2025-03-30
>
> We would like to thank the reviewer. The comments and questions are relevant and will help us improve the paper.
>
> # Previously published references
>  - We would like to highlight that important references for this work have been published. Specifically, Chamma et al. 2023 in NeurIPS and Verdinelli & Wasserman, 2023 in Statistical Science. We will update the bibliography to include the final published version of the Verdinelli & Wasserman paper instead of the preprint.
>
>  # Connection between CATE and variable importance
>   - Following the distinction made by Verdinelli & Wasserman, 2023, our work focuses on population-level variable importance. Estimating how important a particular feature is in the underlying data-generating process $\mu$ is done by estimating the total Sobol index using the LOCO procedure. For a variable $j$, $\Psi_j^* = \mathcal{L}(y, \mu{-j}(X^{-j})) - \mathcal{L}(y, \mu(X))$, where $\mu_{-j}$ is a sub-model estimated without the variable $j$.
>
>
>  - In the present work, the focus on causality is motivated by the type of problem we are interested in: interventional data where patients are assigned to treatment or control groups. In Verdinelli & Wasserman's vocabulary, the underlying data-generating process we wish to study is the individual treatment effect, also known as the conditional average treatment effect (CATE), $\tau(x) = \mathbb{E}[Y(1) - Y(0) | X=x]$.
>
>
>  - Extending the approach from Verdinelli & Wasserman to CATE estimation is not straightforward because the ground truth CATE is not observable — patients are either in the treatment or control group, not both (the fundamental problem of causal inference). In our main paper, we show in Equations 1 and 4 that instead of the loss difference used in Verdinelli & Wasserman, a difference of feasible causal risks can be used to consistently estimate the total Sobol index for the CATE.
>
>  # Motivation and problem setting
>  - The motivation for this approach is to understand, at the population level, what drives heterogeneity in the treatment effect. For example, in a clinical setting, a given treatment might lead to adverse events in a subset of patients (e.g., those with a specific genetic profile or lifestyle ...). The proposed approach would then allow us to identify the risk factors (e.g., genetic profiles or lifestyle ...) associated with a higher risk of adverse events when treated.
>
>
> We appreciate the reviewer's comment and will include this clarification and discussion in the terms of Verdinelli & Wasserman in the final version of the paper.

---

### Official Review · Reviewer_wUZa · 2025-03-18

**Overall Recommendation:** 2

**Summary:**

The paper proposes a new explainability method for the conditional average treatment effect (CATE) estimators, namely, PermuCATE. PermuCATE measures global features importances and is based on a conditional permutation importance (CPI) methods. The authors demonstrated that PermuCATE aims to estimate the expected conditional variance of CATE, given the subset of covariates (total Sobol index). Also, they showed, that under additional assumptions,  PermuCATE achieves lower variance than the existing leave-one-covariate-out (LOCO) method. Ultimately, the authors empirically demonstrate the superiority of their method in terms of statistical power and precision.


**Post-rebuttal response**: I thank the authors for addressing the majority of my concerns. Still, I have two remaining concerns: (1) the presentation and (2) the efficiency of the method.
1. The theoretical results (namely, Propositions  3.4 and 3.5) need to be carefully revisited, especially, considering the large number of corrections and edits during the rebuttal. Here, I also mean streamlining the notation.
2. I read the latest authors' response and now it looks like the proposed method is asymptotically strictly worse than the existing LOCO: the proposed method contains the error of the first order wrt. the estimated $\hat{\tau}$ whereas the  LOCO has the same error squared (= hence, less sensitive to the estimation of $\tau$). Therefore, I don't see how the method with the apriori asymptotically worse performance can be preferred in practice given that we cannot reliably choose or validate Sobol index predictions simply from observational data.

Hence, I keep my current score and encourage authors to further adjust their method so that it achieves the lowest variance (= semi-parametric efficiency bound).

**Claims And Evidence:**

The major claims are supported by proofs and the authors provided an extensive empirical evaluation of their method.

However, I have several major concerns regarding the claimed properties of PermuCATE:
1. I find it hard to believe that the method achieves lower variance than LOCO, given a semi-parametrically efficient variant of LOCO  [1] offers the asymptotically efficient estimator of the target total Sobol index. I encourage the authors to provide a wider discussion on this (also, take into consideration issue 1 in Theoretical Claims, as it seems that Prop. 3.4 is missing an error term).
2. The residuals for conditional perturbations of covariates are estimated on the same subset (D_test) that is being used to estimate the global variable importance. Doesn’t this introduce estimation bias or compromise statistical tests for variable significance?

If all of the issues from above and below could be resolved, I would be happy to raise my score.

References:
- [1] Hines, O., Diaz-Ordaz, K., and Vansteelandt, S. Variable importance measures for heterogeneous causal effects. arXiv, 2022. doi: 10.48550/arxiv.2204.06030.

**Essential References Not Discussed:**

To the best of my knowledge, all the most important works were discussed here.

**Experimental Designs Or Analyses:**

See Methods And Evaluation Criteria.

**Methods And Evaluation Criteria:**

The chosen baseline methods are relevant and make sense. However, the authors did not discuss whether the chosen benchmarks satisfy Assumption 3.1 needed for the method’s consistency. Thus, I would like to see a clear separation between the benchmarks that do and do not satisfy Assumption 3.1.

**Other Comments Or Suggestions:**

- Did you mean “pseudo-outcomes” instead of “potential outcomes” in line 152?
- I would be more precise in lines 114-115 (2nd column) by defining what “comparable risks” mean. It is important to mention, that both the DR-learner risk (=PO-risk) and R-learner risk yield the same CATE estimators **only** in population and when the ground-truth nuisance functions are used [1].
- I had a hard time distinguishing the setting and related work. I suggest the authors split Sec. 2 into two parts for better readability.
- Some notation is not properly defined or introduced (e.g., $\beta_j$ in Eq. 6 or $\Psi_{\text{LOCO}}$ in Eq. 8.

References:
- [1] Morzywolek, Pawel, Johan Decruyenaere, and Stijn Vansteelandt. "On a general class of orthogonal learners for the estimation of heterogeneous treatment effects." arXiv preprint arXiv:2303.12687 (2023).

**Other Strengths And Weaknesses:**

Nothing new here.

**Questions For Authors:**

1. Is there a reason why the risk difference (line 13 of Alg. 1) is divided by 2? It seems like this only complicates the derivations.

**Relation To Broader Scientific Literature:**

The paper suggests a lower variance alternative to the existing CATE variable importance method, LOCO [1]. Yet, this comes at a cost of the additional assumption (Assumption 3.1), namely the additivity of noise for conditional distributions of covariates. In my opinion, this is a pretty strong assumption (although the authors argued differently in Remark 3.3), which limits the application of PermuCATE in practice.

References:
- [1] Hines, O., Diaz-Ordaz, K., and Vansteelandt, S. Variable importance measures for heterogeneous causal effects. arXiv, 2022. doi: 10.48550/arxiv.2204.06030.

**Theoretical Claims:**

I found several inaccuracies in the theoretical statements:
1. In Prop. 3.4, it is hard to believe that the bias of estimating the total Sobol index does not depend on the error of estimating the ground truth CATE, namely, $O_P(\lVert \tau - \hat{\tau} \rVert^2)$. Upon checking the proof in the Appendix, it seems that indeed, the authors missed this term while transitioning between the proofs of Prop. 3.2 and 3.4.
2. I don’t think the additive noise assumption is necessary for the outcome, as introduced in Appendix A.2 (line 575). The general DR-/R-learners do not make such an assumption [1].

References:
- [1] Morzywolek, Pawel, Johan Decruyenaere, and Stijn Vansteelandt. "On a general class of orthogonal learners for the estimation of heterogeneous treatment effects." arXiv preprint arXiv:2303.12687 (2023).

---

> ### Author Rebuttal · Authors · 2025-03-30
>
> We thank the reviewer for the thorough review. The comments and questions are relevant and will help us improve the paper.
> # Dependence of PermuCATE on the estimate of $\tau$
>  - To clarify the dependence of PermuCATE on the estimation of $\tau$, we first provide some intuition: LOCO and PermuCATE quantities are estimated by computing a difference between two risks (eq 1 and 4). Contrary to LOCO, PermuCATE uses the same estimator ($\hat{\tau}$) in both terms of the difference, leading to estimation terms that cancel each other out. This intuition is demonstrated in Supplement A.4. Specifically, in L715, the second-order $\tau$ estimation term cancels out using that $\\mathbb{E}[(\tau (X_{P,j})-\hat{\tau}(X_{P,j}))^2|\mathcal{D_{train}}]=\mathbb{E}[(\tau (X)-\hat{\tau}(X))^2|\mathcal{D_{train}}]$ because by construction $X_{P,j}\overset{i.i.d.}{\sim}X$.
>  - The only term involving the estimation error is term C (L697), a first-order error term: $O(\tau-\hat\tau)$. We initially argued that it is negligible compared to second-order terms and proposed to ignore it. However, we agree that clarifying the dependence on the estimation error of $\tau$ is important and will rephrase proposition 3.4 accordingly.
>  - The Lipschitz assumption aimed to show that in equation 9, the bias term carries the error in estimating the conditional distribution $\nu_j$. As confirmed by experiments, this assumption is not restrictive.
>
> To clarify these last two points, we propose to rephrase the proposition using the intermediate step (eq 9) of the proof in A.4.
>
> ## Proposition 3.4:
> Under Assumption 3.1, for a consistent CATE estimator $\hat{\tau}$, the finite sample estimation of the importance for the $j^{\text{th}}$ covariate is:
> $$\frac{1}{2}\mathbb{E}[\widehat{\Psi_{\mathrm{CPI}}^j}|\mathcal{D_{train}}] - \Psi_j^* = \mathbb{E}[(\hat{\tau}(X_{P,j}) - \hat{\tau}(\widehat{X_{P,j}}))^2| \mathcal{D_{train}}] + O_P(\tau - \hat{\tau})$$
>
> **Remark**: While the first term involves the CATE estimate $\hat{\tau}$, it does not involve the estimation error.
>
> ## Corollary:
> Under the additional assumption that $\hat{\tau}$ is Lipschitz continuous and $\hat\nu$ is consistent, the right term becomes $$O_P(||\nu_{j} - \widehat{\nu}_{j}||^2) + O_P(\tau - \hat{\tau})$$
>
> # Variance comparison between LOCO and PermuCATE
>  - Indeed, LOCO is asymptotically efficient. However, please note that this convergence rate explains its asymptotic behavior and relies on the convergence rates of the models used. In finite samples, the estimation error of these models drives the variance of PermuCATE and LOCO. As shown in eq. 5 and 7, PermuCATE has a smaller dependence on the complex model $\tau$ and, therefore, a smaller finite sample variance. This was experimentally confirmed in Figure 2 and the additional simulation study mentioned below.
>
> # Subset used for conditional perturbations
>  - The goal of the conditional perturbation is to sample from the conditional distribution $X_{P,j} \sim p(X^j|X^{-j})$. Under assumption 3.1, each covariate can be decomposed as a function of the other covariates plus an additive noise term $X^j =\nu_j(X^{-j}) + \epsilon_j$. As presented in algorithm 1, the function $\nu_j$ is estimated on the training data. This avoids overfitting, which could lead to perfectly predicting $X^j$, not perturbing the data and leading to vanishing importance. Then, to sample the additive noise term, we use permutations of the residuals: $\epsilon_j = \mathrm{shuffle(X^j - \nu_j(X^{-j}))}$. We would like to insist that no parameter estimation is involved in the permutation of the residuals and, therefore, no information leakage from the test set. This also maintains scikit-learn API compatibility by using the `fit` method (with training data) to estimate $\nu_j$, and the `score` method (with test data) to sample perturbations.
> # Benchmarks that do and do not satisfy Assumption 3.1
>  -  The simulation scenarios LD, HL, and HP (see datasets in the main paper) used multivariate Gaussians, which satisfied this assumption.
>  - We refer the reviewer to the response provided to Reviewer jT2T, section "Complexity of the conditional distribution estimation." We clarify that the IHDP benchmark uses real-world data, where this assumption cannot be verified. Furthermore, we provide an additional experiment where the dependency structure of the covariates is more complex than that of multivariate Gaussians. In both cases, PermuCATE outperforms LOCO, supporting the validity of assumption 3.1.
>  - We also refer to the section "Discussion of model-x knockoff" in response to Reviewer jT2T, where we discuss that this assumption is common in the literature and review references. This assumption ensures valid conditional sampling from $p(X^j|X^{-j})$. However, any other sampling method could be used as a drop-in replacement.
>
> # Factor 2 in risk difference
>  - As demonstrated in Proposition 3.2, PermuCATE estimates the total Sobol index up to a factor of 2.

---

> > ### Comment · Reviewer_wUZa · 2025-04-04
> >
> > Thank you for the response and for correcting some of the mentioned issues.
> >
> > I have a follow-up question regarding "Dependence of PermuCATE on the estimate of $\tau$". Now as you incorporated the error of estimating $\tau$ into the error of estimating the Sobol index, it looks like this error is of the same order. At the same, time this error term is squared for LOCO (according to Proposition 3.5), which suggests LOCO is more robust than the proposed method.
> >
> > Also, my comments & suggestions remained unanswered. Thus, I tend to retain my current score.

---

> > > ### Author Response · Authors · 2025-04-07
> > >
> > > # Follow-up on the dependence of PermuCATE on the estimate of $\tau$
> > > Thank you for following up. We would like to clarify some imprecisions in our rebuttal that may have led to confusion.
> > > - Specifically, we would like to address the error term in PermuCATE, which comes from term C in L697 of the manuscript. This term is more precisely $O_P(\mathbb{E}_{X\sim D_test}[\tau(X) - \hat\tau (X)])$, where the expectation is taken over the test set and $\hat\tau$ is estimated on the training set.
> > >
> > > - We want to clarify the terminology used. When we referred to the "first-order" term in PermuCATE, we meant the "linear term," which corresponds to the mean bias, using the linearity of the expectation. For example, in a linear model with centered (X), this term will always be 0. By "second-order," we meant the "quadratic term" to describe the MSE term in Proposition 3.5 for LOCO. When limited training data is available, this generalization error term is large, contributing to the increased variance of LOCO, as shown in Figure 2 (small (n) on the left of the x-axis).
> > >
> > > - Moreover, we did not claim that PermuCATE achieves asymptotic semi-parametric efficiency. Our experiments show that in certain scenarios, the variance of LOCO decreases faster asymptotically (see Figures 2a and 2e), which can be attributed to the MSE convergence rate. Our analysis focuses on the finite (training) sample regime, motivated by the scarcity of interventional data. In this non-asymptotic context, these estimation error terms explain the larger variance observed with LOCO, particularly with misspecified models, such as Ridge in non-linear scenarios (Figure 2b) or deep learning models (Figures 2d and 2f).
> > >
> > >
> > > To conclude, in regimes with limited training data, which are common in interventional studies, the MSE term in LOCO becomes particularly problematic, leading to higher variance. Conversely, PermuCATE exhibits greater robustness in such contexts. While asymptotically, the variance of LOCO may decrease faster, this behavior is noticeable in data regimes where both methods benefit from sufficient statistical power (see Figure 3).
> > >
> > > We realize that our initial wording in the rebuttal did not accurately describe the proof provided in A4. We hope this clarification addresses the reviewer's concerns.
> > >
> > >
> > > # Comments & suggestion
> > > We apologize for not addressing all points in our initial rebuttal. The 5000-character limit forced us to make decisions and selectively address comments. We are glad to use the additional characters here to address all the points raised:
> > >
> > >  - **L152 correction**: We indeed meant to refer to pseudo-outcomes. We will correct this in the revised manuscript.
> > >  - **Equivalence of Risks**: We agree that the risks are equivalent in the population when using the true nuisance functions. In the main paper, we used the term "directly comparable" to refer to the result in Appendix A2, which shows that the oracle PEHE appears directly in the decomposition of the PO-risk, whereas it appears in a re-weighted form in the R-risk.
> > >  - **Related Work subsection**: To improve the readability of Section 2, we plan to add a clear separation and subsection title: Related Work.
> > >  - **Notation Consistency**: To clarify notation consistency, we will add a sentence to explain that $\beta_j$ refers to the $j^{th}$ coefficient of the vector $\beta$, while $\Psi_{LOCO}^j$ corresponds to the importance of the $j^{th}$ variable measured using the LOCO approach.

---

### Official Review · Reviewer_jT2T · 2025-03-25

**Overall Recommendation:** 3

**Summary:**

The paper proposes a variable importance measure (VIM) to understand the variables driving the conditional average treatment effect (CATE) function. The measure is based on the principle of conditional permutation importance where the variable of interest is permuted while keeping matching its conditional distribution and then checking the impact on CATE through a quantity known as total Sobol index. The work studies the estimation error, finite-sample variance, and type-I and II error of the proposed VIM to an existing measure named LOCO from Hines et al. 2022. Main contributions are to develop a natural permutation-based VIM and to show that it performs favorably in extensive experiments.

### update after rebuttal

I have read the response and other reviews. The response clarifies my concerns on interpretation of the Sobol index and performance of the method for more complex covariate distributions. Thanks for including an additional experiment.

However, the theoretical claims on estimation errors/variance for LOCO vs proposed method require substantive changes based on the discussion with Reviewer wUZa. Asymptotically, LOCO seems to be better whereas in finite data experiments proposed methods is consistently better. The authors provide an hypothesis for why finite sample performance could be better, namely, MSE error in tau is worse. I would suggest the authors to carefully check the hypothesis in experiments and thus provide an explanation grounded in their theoretical analysis.

More importantly, choosing between LOCO and proposed method is not straight-forward, a concern that has not been addressed. Authors should discuss tests for the assumption or procedures to choose between the two methods. More examples of settings where the data might favor LOCO will help.

Given the experimental results on the proposed method, I am positive that the work is significant. The theoretical justification and guidance for using the method could be improved. Hence, I raise my score to 3 weak accept.

**Claims And Evidence:**

- Claims on inferential properties of the VIM are demonstrated convincingly by experiments.

- Claims on benefits of the proposed method are not very rigorous. The comparison of bias between LOCO and PermuCATE in eps (5) and (7) is a bit misleading. Proposition 3.4 hides the fact that PermuCATE also depends on how good is the CATE estimate tau since it is assumed to be consistent and Lipschitz. Therefore, both methods are susceptible to estimation errors in tau. The eps (5) and (7) hide dependence on sample sizes and assume consistency for tau. Therefore, it is misleading to say they are finite sample analyses. Relatedly, eq (7) does not clearly point to a worse dependence on dimension of covariates. The claim in line 342 is not supported by the theoretical results. Both (5) and (7) will depend on dimension.

**Essential References Not Discussed:**

The paper can discuss other methods that are based on the principle of leave one covariate out such as Zhang and Janson 2020 and more papers on VIMs based on model-x knock off if any.

Lu Zhang, Lucas Janson. Floodgate: inference for model-free variable importance. arXiv 2020 https://arxiv.org/abs/2007.01283

**Experimental Designs Or Analyses:**

- I checked the design for synthetic experiments which control for complexity of conditional outcome function.

- The complexity of conditional covariate density function is not tested adequately. Since Assumption 3.1 on covariate density is important to the results, I think the experiments need to vary complexity of density function as a way to check sensitivity to the assumption, including settings when the assumption is not true.

**Methods And Evaluation Criteria:**

- The permutation-based variable importance is a natural concept and the method skillfully develops it for CATE.

- Experiment setup including the metrics and baselines are relevant.

**Other Comments Or Suggestions:**

Minor comments, no response is requested for the following

- Some terms in the introduction were not introduced. Please describe what is the purpose of variable importance measures, in what way they help in interpretation, feasible risk, and what is the outcome or effect in CATE in the biomedical applications discussed in introduction. Explain how showing variable importance for CATE will help in biomedical applications.

- Please discuss CPI in some more detail in related work or introduction to make the reader appreciate its importance.

- Please explain the remark on total Sobol index in line 135.

- Consider labeling Eq. (3) as R(…)=…

- I really like the Figure 1. It conveys information on effect of sample size on VIM, comparison to true value, and hypothesis test results quite clearly.

- CPI and PermuCATE are used interchangeably, CPI in notation and PermuCATE in text. Please consider using the same name everywhere.

- Please define support in line 370. Does it mean variables used in the true outcome function?

- Please complete the sentences at the end of Remark 3.3. It is evident they support the claim, however, can be written more explicitly.

- Motivation for the method is from high-dimensional settings like SNPs data whereas the IHDP datasets in evaluation study has moderate number of covariates.



Typos
Line 066 an
Line 162 estimated
Line 333 pseudo

**Other Strengths And Weaknesses:**

# Strengths

- Method is conceptually simple and inherits the statistical inference guarantees on type-I error from prior results.

- Proposed method provides a useful alternative to existing variable importance measures by leaning more towards modeling the conditional covariate distribution instead of the conditional outcome distribution. Therefore, the method might be more suitable in some applications.

- Experimental validation is thorough. Authors rigorously estimate the importance measures by using flexible CATE-learners wrapped in ensemble methods like super learner and use cross-fitting to reduce variance. Throughout the results report confidence intervals or p-values over sufficiently many repeated samples of the data.


# Weaknesses

- It is unclear when to use LOCO vs PermuCATE since they have similar statistical inference properties but differ in assumptions. Consider discussing a test for checking Remark 3.3 that states that the conditional covariates are easier to model than conditional outcome function. Was it the case for IHDP dataset? The claim that modeling conditional covariate distributions are easier for some applications should be more carefully demonstrated or discussed through citing literature.

- The presentation can be improved. I felt that CPI could be explained in more detail before the methods to give the context for comparisons to LOCO. Please present the statistical guarantees for CPI. Please introduce total Sobol index and discuss why it is a good measure for CATE.

- Compared to existing method LOCO, the proposed method is not readily extensible to computing variable importance for variable subsets. The nuisance parameter \nu will be a challenging multivariate regression, whereas LOCO still requires a regression on univariate response. Computing importance for subsets is important to handle highly-correlated variables and to limit the computation since often variable subsets can be grouped together and treated as one variable.

- Experiments on IHDP data did not test whether variables are ranked in the true order. AUC only tests for ranking between important and not important variables. A more prevalent use of variable importance measures is to identify which of the important variables are the most important. Please consider reporting metrics like precision at k or Kendall tau.

**Questions For Authors:**

- Please discuss validity of the claims on variance of the two methods.

- Please intuitively describe total Sobol index.

- Please provide more evidence either from literature or IHDP data that covariate density is easier to model or suggest when to apply the proposed method.

- Please clarify whether the experiment setup checks Assumption 3.1 systematically.

**Relation To Broader Scientific Literature:**

- The method is new and experimental comparison of finite-sample behavior against LOCO are impressive.

- Please discuss the relation to model-x knock off literature in more detail since the methods similarly model the covariate density. Does there exist CATE importance measure from the literature that are natural baselines?

**Theoretical Claims:**

I read proof for Proposition 3.2 carefully and skimmed the proofs for Proposition 3.4 and 3.5. For 3.2, the cited result from Reyero Lobo et al. 2025 shows convergence in Wasserstein which is written as an exact equality in the proof. I think the main result is ok, but the statement should be written carefully.

---

> ### Author Rebuttal · Authors · 2025-03-30
>
> We thank the reviewer for the thorough review. The comments and questions are relevant and will help us improve the paper.
> # Dependence of PermuCATE on the estimate of $\tau$
> - We refer the reviewer to the response provided to Reviewer wUZa, in the section "Dependence of PermuCATE on the estimate of $\tau$ ". We explain why second-order error terms cancel out in the estimation of PermuCATE, and propose to rephrase proposition 3.4 to include first-order error terms.
>  - We agree with the comment that eq 7 does not explicitly point to a worse dependence on the dimension. We plan on rephrasing the L342 to clarify that this is an empirical observation.
> # Complexity of the conditional distribution estimation
>  - The IHDP benchmark uses real-world data, with non-Guassian continuous features and imbalanced categorical (up to 4 categories) features. Because the ground truth for the conditional distribution is not available, the validity of Assumption 3.1 cannot be assessed. However, Figure 4 shows that PermuCATE outperforms LOCO with better statistical power and AUC for detecting the important features while controlling the type-1 error on this dataset.
>  - In addition to this benchmark, we propose an additional experiment where we revisit the simulation study presented in Figure 2 also to vary the complexity of the conditional distributions. To do so we use simulations inspired by [1], we sample the covariates using a latent variable model. We first sample latent variables from mixtures of Gaussians and then generate observed covariates $X$ through a non-linear transformation with interaction terms of the latent variables. This results in non-Gaussian covariates with complex conditional distributions, illustrated in this figure: https://imgur.com/a/JqT04in In a), presents the marginal distributions, b) the kurtosis of generated variables (blue) compared with the kurtosis for Gaussians (orange) for comparison, and c) their correlation matrix. Similarly to Figure 2, we generate 2 CATE functions, one linear and one non-linear and compared the variance of PermuCATE and LOCO for three different models at varying sample sizes. The results shown in this figure: https://imgur.com/a/t0qhb61 reveal a lower variance for PermuCATE compared to LOCO similar to Figure 2. Our interpretation is that the complexity of covariates distributions also affects LOCO, by making the CATE harder to estimate.
>
> [1] Hollmann et al. Accurate predictions on small data with a tabular foundation model. Nature 2025
> # Description of the total Sobol index
>  - We refer the reviewer to the response provided to Reviewer VGy6, in the section "Description of the total Sobol index". We provide an intuitive presentation of the total Sobol index. We will integrate this presentation into the final version of the paper.
> # Discussion of model-x knockoff
> The proposed approach shares similarities with model-X knockoffs (KO) as both model the conditional distribution of covariates, assuming this task is easier than estimating the quantity of interest (e.g., CATE). This point is discussed in Remark 3.3 of the main paper. A key difference is that model-X KO is designed for variable selection, whereas PermuCATE and LOCO focus on variable importance estimation. The latter provides richer information by measuring importance (total Sobol index) rather than just binary selection. Additionally, KO handles multiple testing, while our method ensures type-I error control. Also, constructing KO variables requires complete exchangeability, a stricter condition than the conditional independence needed for CPI. The conditional randomization test (CRT) from Candès et al. 2018 is more comparable to our approaches for individual conditional independence testing but is much more computationally expensive.
>
> Besides the work from Hines et al. 2020, we are unaware of other methods that provide model agnostic variable importance for CATE.
> # Extension to groups of variables
>  - As mentioned in L235 and demonstrated in Appendix A.12, CPI-based approaches can be extended to groups of variables. CPI with grouping in the context of prediction problems was formally treated in https://doi.org/10.1609/aaai.v38i10.28997. There is no particular complexity when performing multi-output regression. For each variable of the group, the conditional distribution can be estimated independently and in parallel: $\forall j\in G$ estimate $\nu_j = \mathbb{E}[X^j|X^{-G}]$, with $X^{-G}$ the complementary of group G.
> # Ranking of important variables
>  - The ranking of important variables is indeed richer information than the AUC for binary classification of important variables. However, except for simple scenarios (Gaussian covariates and linear CATE) such as those presented in Figure 1, the ground truth importance (or ranking) is not known. For IHDP, two problems prevent the analytical computation of the total Sobol index: the true conditional distribution is unknown, and the CATE function is non-linear.

---

### Decision · Program_Chairs · 2025-05-01

**Decision:**

Accept (poster)

**Comment:**

This paper focuses on measuring variable importance in conditional average treatment effect (CATE) functions. Existing methods such as LOCO are applied to CATE estimators, and the authors propose the application of CPI as an alternative. A novel method, PermuCATE, based on CPI, is introduced to measure global feature importance in CATE estimators. Empirically, PermuCATE demonstrates superiority over LOCO in terms of faster convergence, lower variance, statistical power, and precision.

Overall, the reviewers' feedback was positive. The method was commended for being conceptually simple and inheriting the statistical inference guarantees on type - I error from previous results. Most of the major concerns raised by reviewers, such as enhancing theoretical justification and providing explanations for the connections with related work, have been addressed.

After reviewing the paper, the reviewers' comments, and the authors' rebuttal, I recommend the acceptance of this paper at this time.